# Transient reactivation of small ensembles of adult-born neurons during REM sleep supports memory consolidation in mice

Sakthivel Srinivasan[1,5], Iyo Koyanagi[1,2,5], Pablo Vergara [1], Yuteng Wang[1,2], Akinobu Ohba[1], Toshie Naoi[1,2], Kaspar E. Vogt [1,2], Yoan Chérasse [1,2,3], Noriki Kutsumura [1,2], Takeshi Sakurai [1,2,3], Taro Tezuka[4] & Masanori Sakaguchi [1,2,3] ✉

While memory consolidation is widely believed to require memory reactivation synchronized with theta oscillations during REM sleep, direct causal evidence linking specific neuronal ensembles to this process has been lacking. Strong theta oscillations arise in the hippocampal dentate gyrus, where a small population of principal neurons is continuously generated throughout adulthood. Although these adult-born neurons (ABNs) are known to modulate hippocampal circuits for memory, the causality between their specific information content and memory-related behavior was unknown. Here, we show that REM sleep reactivation of memory ensembles comprising as few as ~3 ABNs is necessary for fear memory consolidation. Crucially, we found that the synchronization of ABN activity with a specific theta phase is essential for associative memory consolidation. Our findings thus provide causal evidence that consolidation critically depends on both the reactivation of minimal neuronal populations and precise neuronal coordination within theta-defined temporal windows.

During rapid eye movement (REM) sleep, neuronal ensemble activity related to behavioural experiences is reproduced in the hippocampus and synchronised with a specific phase of theta oscillation[1,2]. However, causal evidence linking the reactivation of these neuronal ensembles to memory consolidation during REM sleep is lacking. Previous studies demonstrate that theta oscillations during REM sleep are essential for the consolidation of fear memory[3]. Prominent theta oscillations are observed in the dentate gyrus (DG) of the hippocampus during REM sleep[4]. DG neurons synchronise their activities with theta oscillations during REM sleep[5] and encode fear memory[6]. Principal DG excitatory neurons, known as granular neurons (GNs), include neurons born in adulthood[7,8]. A subpopulation of these adult-born neurons (ABNs) that are active during fear learning reactivate during subsequent REM sleep, and silencing the overall activity of ABNs during REM sleep impairs fear memory consolidation[9]. In this study, we show causal evidence linking the reactivation of neuronal ensembles to memory consolidation during REM sleep.

## Results

### ABN ensemble reactivation in REM sleep

To examine the activities of ABN ensembles in freely moving, naturally sleeping mice, we performed calcium (Ca²⁺) imaging using a miniaturised microendoscope. To accomplish this, we induced expression of a genetic Ca²⁺ sensor, GCaMP6s, in a triple-transgenic mouse line, pNestin-CreER^T2/cfos-tTA/TRE-LSL-GCaMP6s(nestin/cfos/ G6s) (Fig. 1a)[10,11]. GCaMP6s was expressed under the promoter of an

[1]International Institute for Integrative Sleep Medicine (WPI-IIIS), University of Tsukuba, Tsukuba, Ibaraki, Japan. [2]Tsukuba Institute for Advanced Research (TIAR), University of Tsukuba, Tsukuba, Ibaraki, Japan. [3]Institute of Medicine, University of Tsukuba, Tsukuba, Ibaraki, Japan. [4]Institute of Systems and Information Engineering, University of Tsukuba, Tsukuba, Ibaraki, Japan. [5]These authors contributed equally: Sakthivel Srinivasan, Iyo Koyanagi. ✉e-mail: sakaguchi.masa.fp@alumni.tsukuba.ac.jp

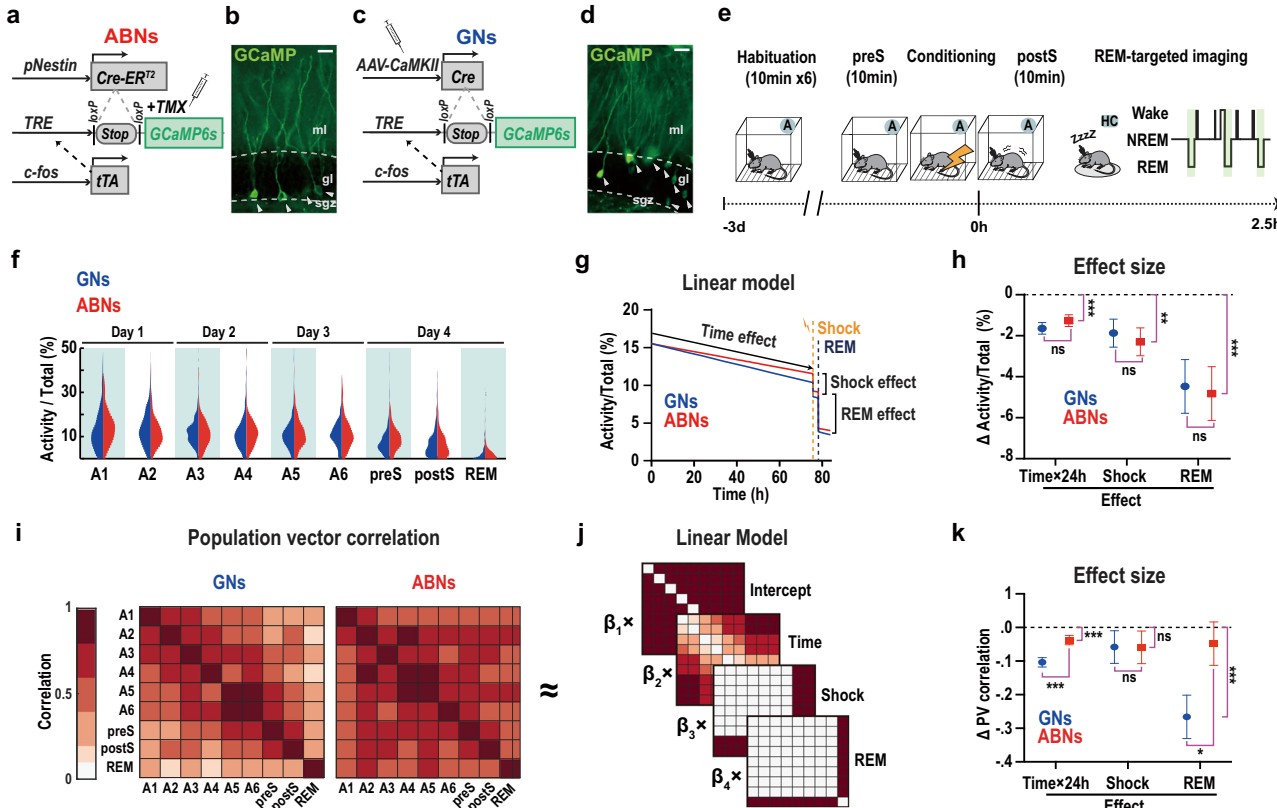

**Fig. 1 | Ca²⁺ activity of active ABNs and GNs during conditioning and sleep.**
**a**, **c** Transgenic method for labelling ABNs and GNs. **b**, **d** Representative image of GCaMP6s expression (arrow heads). Similar expression was observed over 3 mice in each condition. Scale bar, 25 μm. **e** Protocol for Ca²⁺ imaging during conditioning (preS, pre-shock; postS, post-shock) and sleep. **f** Distribution of the average activity of individual ABNs and GNs as a percentage of total activity across sessions. **g** Linear model of average neuronal activity as function of time, shock exposure, and REM sleep. **h** Estimated effect size for each variable. Time×24h: 24-h activity changes over time. Data are presented as mean values and standard errors of the regression coefficients. See Supplementary Table 1 for the statistical details. **i** Correlation matrices of PVs across sessions. **j** PV correlations were modelled using a linear

combination of matrices reflecting the baseline correlation (intercept), time, shock exposure, and REM sleep. **k** Effect size for each variable (i.e., $\beta$ in (j)). ABN PVs were more temporally stable and exhibited stronger correlations between context A and REM sleep than GN PVs. Data are presented as mean values and standard errors of the regression coefficients. See Supplementary Table 2 for the statistical details. For all panels: GNs, $n = 363$ neurons in 3 mice; ABNs, $n = 136$ neurons in 3 mice. ABNs, adult-born neurons; TMX, tamoxifen; mol, molecular layer; gl, granular cell layer; sgz, subgranular zone; GNs, granular neurons; A, context A; HC, home cage; Wake, wakefulness; NREM, Non-rapid eye movement sleep; REM, REM sleep; PV, population vector; ns, not significant; **, $p < 0.01$; ***, $p < 0.001$.

immediate early gene, *cfos*, to induce GCaMP6s expression in differentiated ABNs and avoid possible complications of expression in progenitor cells[9]. Consistent with a previous report[9], 4 weeks after tamoxifen (TMX) injection, adult mice showed GCaMP6s expression confined to the DG subgranular zone, indicating its expression in ABNs (Fig. 1b). We focused on these 4-week-old (young) ABNs because their overall activity is necessary for memory consolidation during REM sleep[9]. For comparison, we also examined developmentally-born GNs by injecting an adeno-associated viral vector carrying *pCaMKII-Cre* genes into the outer half of the granular layer of the cfos/G6s mouse DG (camk/cfos/G6s; Fig. 1c-d), where ABNs are scarce[7].

To analyse the activity of ABN ensembles related to behavioural experiences, we employed a classical fear conditioning paradigm (Fig. 1e), which has been shown to require REM sleep for memory consolidation[3,9,12]. This paradigm allowed us to separately analyse the encoding of a conditioned stimulus (context A) and its association with an unconditioned stimulus (foot shock). We first exposed mice to context A (Supplementary Fig. 1) six times over 3 days (10 min per session, twice per day). On day 4, mice were exposed to context A (pre-shock period, 10 min) followed by delivery of foot shocks. After the post-shock period (10 min), mice were returned to their home cage. We performed Ca²⁺ imaging only during context A exposures and subsequent REM sleep episodes within the memory consolidation period in the home cage to avoid potential photobleaching of

GCaMP6s by continuous light exposure (Supplementary Movie 1). We tracked the same neurons across recording sessions[13]. The activity can be influenced by the loss of novelty as a function of time[14–16], shock exposure during fear conditioning[17], and REM sleep[9]. As these processes occurred simultaneously throughout our experimental paradigm, it is challenging to isolate and identify their individual contributions. Therefore, we used a linear model to simultaneously incorporate these effects and model average activity (Fig. 1f-g). We found no difference in ABN and GN activity across recording sessions (Fig. 1h, Supplementary Table 1). Consistent with previous reports[9,14,17], both neuron types showed a cumulative decrease in activity over time, following shock exposure, and during REM sleep (Fig. 1h).

To examine the reactivation of ensemble activities representing memory during conditioning and REM sleep, we calculated correlations between population vector (PV) activities in different session pairs (Fig. 1i). Using a linear model, we evaluated whether PV correlations changed in response to time, shock, or REM sleep (Fig. 1j). Although both ABN and GN PV drifted over time, ABN PV were more stable over time than GN and had stronger PV correlations between context A (i.e., A1-A6, preS, postS) and REM sleep (Fig. 1k, Supplementary Table 2). These results were not explained by differences between ABNs and GNs in data dimensionality (Supplementary Fig. 2a), the number of neurons (Supplementary Fig. 2b-c, Supplementary Table 4), or duration of REM sleep (Supplementary Fig. 2d), nor by the

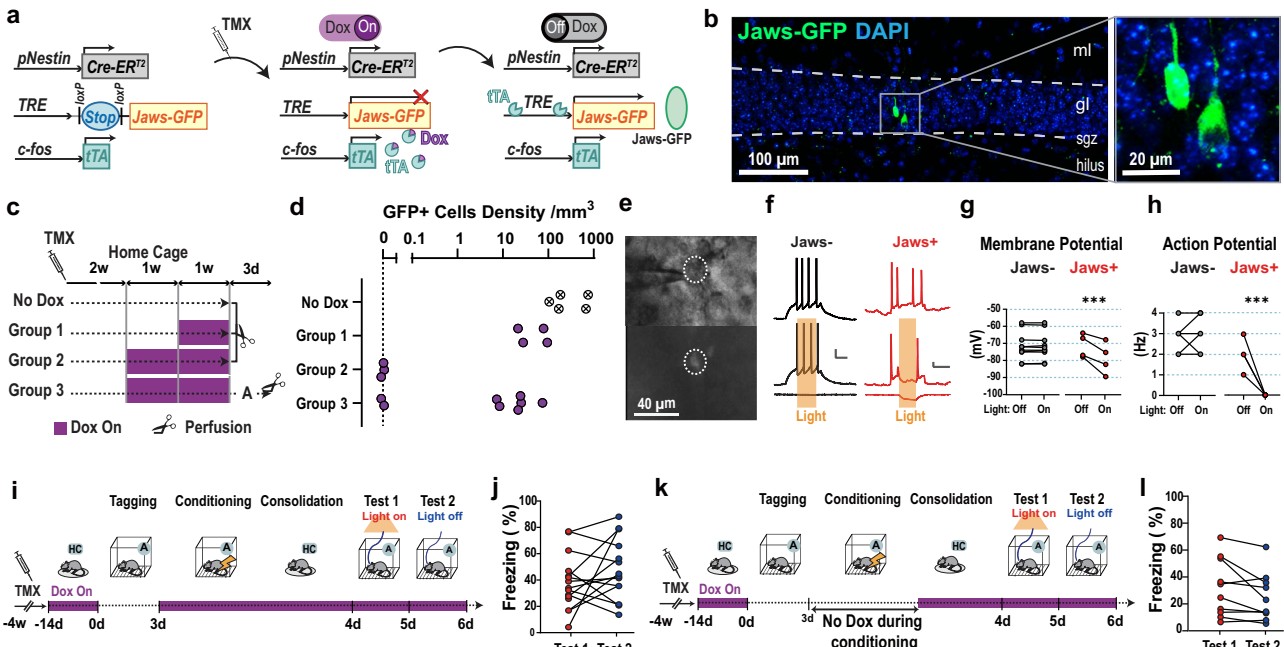

**Fig. 2 | Silencing tagged ABNs during memory retrieval does not impair memory. a** Transgenic method for tagging active ABNs in a temporally specific and reversible manner. **b** Visualization of Jaws-GFP in young ABNs using immunostaining, which enhanced the somatic signal (brain section from Group 3 in (**c**); ml, molecular layer; gl, granular cell layer; sgz, subgranular zone). Similar expression was observed in 6 mice. **c** Dox dose- and experience-dependent analysis of Jaws-GFP expression. **d** Jaws-GFP cell density (No Dox, $n = 5$ mice; Groups 1-2, $n = 4$; Group 3, $n = 8$). **e** Jaws-GFP expression in the DG in acute brain slices. The experiment was independently repeated 4 times. **f–h** Effect of light delivery on resting membrane potential and electrically evoked action potentials in Jaws- ($n = 10$) and Jaws+ ($n = 4$) neurons. Scale bar: 10 mV and 100 ms. Two-way repeated measures ANOVA with Sidak's multiple comparisons. For (**g**), Jaws-, $p = 0.99$; Jaws +, $p < 0.0001$. For (**h**), Jaws-, $p = 0.86$; Jaws +, $p < 0.0001$. **i**, **k** Protocol for silencing tagged ABNs during memory retrieval. **j**, **l** Freezing during memory retrieval test. Two-tailed paired $t$-test. For (**j**), $n = 14$ mice, $p = 0.21$. For (**l**), $n = 10$, $p = 0.056$. TMX, tamoxifen; Dox, doxycycline; A, context A; HC, home cage; ***$p < 0.001$.

shorter duration of REM sleep sessions compared with other imaging sessions (Supplementary Fig. 2e). These findings suggest that, as a whole, ABN ensembles exhibit a more consistent activity profile across learning and REM sleep, whereas GNs display greater heterogeneity in their involvement during these stages.

## Manipulation of reactivated ABN ensembles

We next examined the causality of ABN memory ensemble reactivation during REM sleep for fear memory consolidation. To genetically tag ABN ensembles that are active during context exposure and examine their function via optogenetic silencing using Jaws opsin[18], we created a nestin/cfos/TRE-LSL-JawsGFP mouse line (nestin/cfos/jaws; Fig. 2a). As expected, Jaws-GFP expression was observed in ABNs in nestin/cfos/jaws (cfos + ) mice (Fig. 2b) but not in nestin/jaws (cfos-) control mice (Supplementary Fig. 3a-c). Jaws-GFP expression in ABNs was suppressed by doxycycline (Dox) in a dose-dependent manner, with complete lack of expression after 2 weeks of Dox-On, confirming its tTA dependency (Fig. 2c-d). Three days of Dox-Off during context A exposures allowed tTA-driven Jaws-GFP expression in a subset of ABNs (Fig. 2c-d). Consistent with previous findings[19], only a few ABNs were tagged in cfos+ (Fig. 2b-d), with a mean (standard deviation) of 2.4 (3.2) neurons per mouse in the light-effective area ($n = 8$ mice; for detailed counting methods, see Methods). The distribution of tagged neurons in the GL is consistent with that in previous studies[20–23] (Supplementary Fig. 3d-e). As expected, tagged neurons were also observed in the olfactory bulb (Supplementary Fig. 3f)[10].

We confirmed that Jaws-expressing ABNs were silenced in a temporally specific and reversible manner by delivering orange light (589-nm laser) to activate Jaws in acute brain slices from nestin/cfos/jaws mice (Fig. 2e-h). We also confirmed that GFP+ neurons showed an ordinal range of input resistance[24–29] (Supplementary Fig. 3g).

When ABNs were depolarised via current injection, orange light transiently decreased Jaws-expressing ABN membrane potential and silenced action potentials with high temporal fidelity and reversibility (Fig. 2e-h). No such responses were observed in non-Jaws-expressing neighbouring cells (Jaws-) (Fig. 2e-h).

Using this system, we investigated whether reactivation of ABN ensembles that were active during context exposures is necessary for memory retrieval. cfos+ mice underwent Dox-On for 2 weeks and were then exposed to context A during a 3-day Dox-Off period to specifically tag context-activated ABN ensembles (this procedure is the same as for Group 3 in Fig. 2c). Mice were then returned to Dox-On to stop further tagging, after which they underwent fear conditioning to form an association between context A and shock (Fig. 2i). We found no difference in contextual fear memory between two retrieval tests in which tagged ABN ensembles were and were not silenced, as evidenced by equivalent freezing behaviour (Fig. 2i-j). Furthermore, when mice underwent context exposure and conditioning without Dox but were treated with Dox immediately afterward, their memory was not impaired when tagged ABNs were silenced during the retrieval tests (Fig. 2k-l). Considering the low overlap between conditioning- and retrieval-activated ABN populations[9,17] (Supplementary Fig. 2f), the reactivation of ABN ensembles representing contextual memory may not play a critical role in memory retrieval, despite the necessity of the overall activity of the ABN population[30,31].

## ABN ensemble reactivation for memory consolidation

Using the same tagging method, we tested whether reactivation of ABN memory ensembles during REM sleep is necessary for memory consolidation. cfos+ and cfos- mice showed similar behavioural responses in the tagging and conditioning periods (Supplementary Fig. 4a-c). Interestingly, delivering orange light specifically during REM

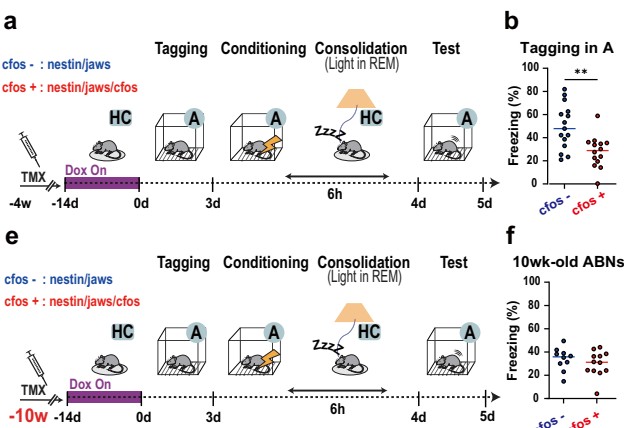

**Fig. 3 | Silencing context A-tagged ABNs during REM sleep impairs memory. a, c, e, g** Protocol for silencing ABNs tagged in context A or C. **b, d, f, h** Freezing during the memory retrieval test. Two-tailed unpaired *t*-test. **b** Tagged in context A: cfos-, *n* = 15 mice; cfos + , *n* = 14; *p* = 0.002. **d** Tagged in context C: cfos-, *n* = 10 mice;

cfos + , *n* = 11; *p* = 0.69. **f** 10-week-old ABNs tagged in context A: cfos-, *n* = 10 mice; cfos + , *n* = 12; *p* = 0.38. **h** GNs: cfos-, *n* = 13 mice; cfos + , *n* = 14; *p* = 0.77. Horizontal bars, mean; A, context A; C, context; HC, home cage; REM, rapid eye movement sleep; **p* < 0.01.

sleep in the memory consolidation period impaired contextual fear memory in cfos+ mice compared with cfos- mice (Fig. 3a-b; Supplementary Fig. 4d, light delivery sensitivity and precision in REM sleep, mean (standard deviation): 0.83 (0.054) and 0.82 (0.12), respectively). Importantly, delivering light during REM sleep did not affect sleep architecture (Supplementary Fig. 4e-i), and the behavioural results were reproduced in an independent experiment with additional rigorous blinding procedures (Supplementary Fig. 5a-b; see details in Methods).

To examine the context specificity of ABN memory ensembles, mice were habituated to a different context, context C (Supplementary Fig. 1; 10 min per exposure, twice a day for a total of six times) during the Dox-Off period and were conditioned and tested in context A (Fig. 3c). Context C exposure resulted in a similar number of tagged ABNs as context A exposure (Supplementary Fig. 5c). Silencing the context C-tagged ABN ensembles during REM sleep did not impair fear memory consolidation (Fig. 3c-d). To eliminate the potentially confounding novelty effect of context A during conditioning[14–16,32], a separate group of mice was habituated to context A with Dox-On, exposed to context C with Dox-Off, and conditioned in context A (Supplementary Fig. 5d). Again, silencing the tagged ABN ensembles during REM sleep did not impair fear memory consolidation (Supplementary Fig. 5d-e). These results indicate that ABN ensembles tagged by context exposures represent context-specific memory, and only the reactivation of ABN ensembles associated with shock during REM sleep is necessary for fear memory consolidation.

Furthermore, no memory impairment was observed when reactivation of context A-tagged fully mature (~10-week-old) ABN (Fig. 3e-f) or developmentally born GN (Fig. 3g-h, Supplementary Fig. 5f-k) ensembles were silenced during REM sleep. Taken together, these results suggest that reactivation of young ABN memory ensembles during REM sleep is necessary for fear memory consolidation.

**ABN theta synchrony for consolidation**

ABNs show highly synchronised ensemble activity with theta oscillations during wakefulness[14], suggesting that ABN memory ensemble reactivations occur during a specific phase of theta oscillations in REM sleep, similar to what is observed in CA1[1]. Therefore, we hypothesised that the ABN ensemble activity synchronised with theta oscillations during REM sleep is necessary for fear memory consolidation. To test this hypothesis, we silenced ABN activity at specific phases of theta oscillations during REM sleep using a closed-loop feedback system and nestin/pCAG-LSL-HalorhodopsinYFP (nestin/halo) mice[9] (Fig. 4a-c).

We employed a trace fear conditioning paradigm, known to be hippocampus- and REM sleep amount-dependent[33–35], in which a temporal gap (i.e., trace interval) is introduced between tone and shock stimuli. This paradigm allowed us to simultaneously assess the involvement of ABNs in both trace fear memory and contextual fear memory consolidation within a single experimental framework. Previously, we used a delayed tone fear conditioning paradigm and demonstrated that ABN activity during REM sleep plays a minimal role in the consolidation of delayed tone fear memory[9], which is hippocampus-independent[36]. Given that the only essential difference between trace fear conditioning and delayed tone fear conditioning is the inclusion of a trace interval, this paradigm is particularly advantageous for evaluating hippocampus-dependent roles of ABNs.

We prepared mouse groups in which young ABN ensemble activity was silenced at one of four phases of each theta oscillation during REM sleep within the memory consolidation period (Fig. 4c-d, Supplementary Fig. 6). Mice in the Phase 1 group, but not Phase 2-4 groups, showed less freezing in both the context and trace fear memory retrieval tests compared with the yoked control group (Fig. 4e). We observed no notable differences between groups in shock reactivity (Supplementary Fig. 7a-b), sleep architecture (Supplementary Fig. 7c-g), or theta oscillations in REM sleep (Supplementary Fig. 7h-i). We further confirmed the theta phase-specific role of ABN activity in memory consolidation by estimating the effect size of silencing ABNs, depending on the timing (i.e., the specific point in the theta oscillation cycle) and the amount of light stimulation (Supplementary Fig. 8). Considering the reactivation of ABN memory ensembles during REM sleep in fear memory consolidation (Fig. 1k) and the necessity of both ABN memory ensemble reactivation (Fig. 3a-b, Supplementary Fig. 5a-b) and the synchrony of ABN ensembles at a specific phase of theta oscillations (Fig. 4e) for fear memory consolidation, these results collectively support the hypothesis that the reactivation of ABN ensemble activities synchronised with local theta oscillations is crucial for fear memory consolidation.

## Discussion

We demonstrated that young ABN memory ensembles reactivate during REM sleep. Furthermore, silencing the reactivation of these ABN ensembles during REM sleep impairs memory consolidation, whereas silencing during a retrieval test during wakefulness does not affect memory. Previous studies report that silencing overall ABN activity impairs memory retrieval[30,31], leading to the hypothesis that ABN ensembles active during learning are transiently recruited into memory circuitry upon encoding salient information. Indeed, ABN activity during fear conditioning is necessary for the memory

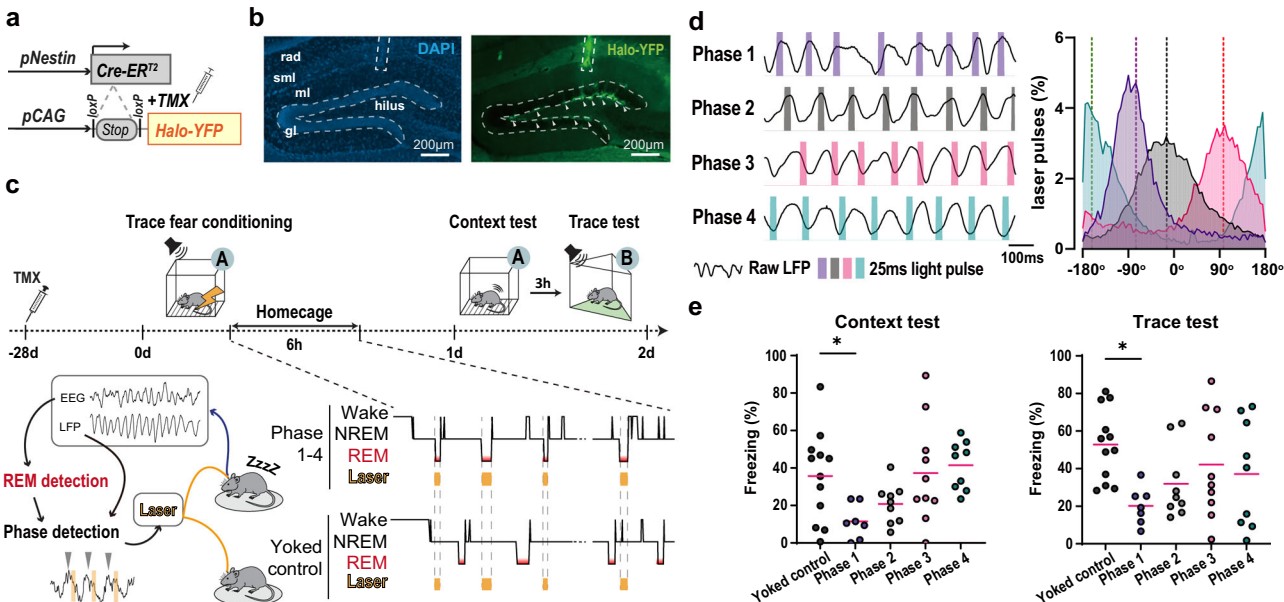

**Fig. 4 | Silencing ABN activity at a specific phase of the local theta oscillation in REM sleep impairs memory. a** Transgenic method for labelling ABNs. **b** Halo-YFP expression in the subgranular layer of the DG. The experiment was independently repeated over 7 times/group. **c** Behavioural procedure for silencing ABNs in a theta phase-specific manner during REM sleep after trace fear conditioning. **d** Left, silencing targets within theta phases. Right, histogram of light delivery in different phase groups. 6 random REM episodes with light stimulation from individual mice

LFP and the laser pulse log were analysed. Phase 1, $n = 7$; Phase 2, $n = 9$; Phase 3, $n = 10$; Phase 4, $n = 9$ mice. Dashed lines, mean phases. **e** Freezing in the context (left) and trace (right) tests. One-way ANOVA with Dunnett's multiple comparisons tests (all vs. yoked control). Yoked control, $n = 12$ mice; Phase 1, $n = 7$; Phase 2, $n = 9$; Phase 3, $n = 10$; Phase 4, $n = 9$ mice. Context test: $p = 0.042$; Trace test: $p = 0.013$. TMX, tamoxifen; context A; B, context B; Wake, wakefulness; NREM, Non-rapid eye movement sleep; REM, REM sleep; Horizontal bars, mean; *, $p < 0.05$.

acquisition[31,37]. Moreover, ABN ensembles that are active during fear conditioning show a gradual decline in activity around 3 to 6 hours after conditioning, as different young ABN ensembles become active during memory retrieval[17]. Therefore, the recruitment of ABN ensembles during associative learning may induce a replacement of the young ABN population encoding a memory after its consolidation. In combination with the tendency of ABNs to respond to novelty[14–16], the transient recruitment of ABNs for memory consolidation and their populational turnover may ensure the efficient association of novel information with salient stimuli.

Regarding our method for labeling ABNs using the triple transgenic mouse line, one might raise concerns about potential bias in labeling specific subsets of neurons due to a reliance on cfos promoter activation. However, activation of the cfos promoter in DG GNs is well-established as a method for labeling cell ensembles that form contextual fear memory engrams[6,38]. Additionally, in our repeated contextual exposure experiments, PV correlations of ABN ensembles remained stable compared with those of GN ensembles (Fig. 1i-k). These findings suggest that our approach reliably and consistently labels functionally relevant ABNs, indicating that any cell-type selection bias associated with cfos promoter activation does not substantially affect the interpretation of our results.

Some mice exhibited no detectable Jaws-GFP+ cells, likely due to individual variability in responsiveness or technical limitations of the cfos promoter-driven expression system (Fig. 2a). Similarly, within the experimental group, some mice exhibited memory consolidation comparable to control mice (Fig. 3b). Due to technical challenges preventing reliable individual-level correlation analyses, we increased the sample size (n > 12 mice per group), replicated the critical experiment with rigorous randomization and blinding (Supplementary Fig. 5a-b), and further confirmed the robustness of results through control experiments, including labeling experiments conducted in a different context (Fig. 3c-d). Overall, our analyses support the conclusion that even a very small number of ABNs can make a major contribution to memory consolidation.

In this study, we demonstrated that the activity of young ABNs at a specific phase of DG theta oscillations during REM sleep, particularly the ascending phase, is essential for the consolidation of fear memory. Periodic theta activity in the DG is generated by rhythmic inputs from the entorhinal cortex[39]. During wakefulness, ABNs exhibit higher synchronization to theta oscillations compared with mature GNs[14]. This enhanced theta synchronization suggests that ABNs play a critical role in efficiently inducing spike timing-dependent plasticity within theta cycles. Indeed, it is well documented that the efficiency of synaptic plasticity induction in the DG depends on theta phase[40]. Young ABNs are susceptible to synaptic plasticity on the input side, owing to their higher membrane resistance[21] and weaker inhibitory inputs[41,42]. Indeed, ABN activity during REM sleep is essential for structural plasticity of their dendritic spines associated with memory consolidation[9].

On the output side, young ABNs exhibit distinct short-term synaptic dynamics compared with mature GNs[24], suggesting their ability to provide effective input to CA3 at specific points within the theta cycle. Based on our LFP recordings near the stratum lacunosum-moleculare, we propose that granule cell layer activity is maximized around the peak following the ascending theta phase. Thus, early firing of ABNs during the ascending phase might enhance DG output at this peak timing. Specifically, high-frequency firing of ABNs could effectively depolarize CA3 neurons[43], thereby preparing CA3 circuits to respond more robustly to subsequent mossy fiber inputs arriving[44]. Furthermore, ABNs may play an important role in finely tuning the balance of excitation and inhibition within theta cycles[39] by potentially adjusting excitatory outputs through interactions with inhibitory networks, thereby facilitating efficient information processing[14,45].

Taken together, these mechanisms suggest that REM sleep-specific synchronization of ABN activity to the ascending phase of theta oscillations effectively promotes synaptic plasticity at both input and output synapses, facilitating neural circuit reorganization critical for memory consolidation. Overall, this study contributes to elucidating the mechanisms by which memory traces are formed during sleep leading to the consolidation of fear memories.

## Methods

### Animals

All animal experiments were approved by the University of Tsukuba Institutional Animal Care and Use Committee (Animal experimental approval# 23-237, Gene recombination experiment approval# 210122). Mice were kept in home cages consisting of a cylindrical Plexiglas container (21.9-cm diameter, 31.6-cm height) in an insulated chamber (45.7 × 50.8 × 85.4 cm) and maintained at an ambient temperature of 23.5 ± 2.0 °C under a 12-h light/dark cycle (9 am to 9 pm) with ad libitum access to food and water in accordance with institutional guidelines. Wild-type C57BL/6J mice, pNestin-CreER[T2] (nestin mice, The Jackson Laboratory, JAX:016261, RRID:IMSR_JAX:016261)[10], Ai94(TITL-GCaMP6s)-ROSA26-ZtTA (TRE-LSL-GCaMP6s) (GCaMP6s mice, The Jackson Laboratory, JAX:024112, RRID:IMSR_JAX:024112), Ai79D(Rosa26-TRE-loxP-stop-loxP-Jaws-GFP)(jaws mice, The Jackson Laboratory, JAX:023529, RRID:IMSR_JAX:023529), and Ai39(Rosa26-pCAG-LSL-eNpH3.0-YFP)(halo mice, The Jackson Laboratory, JAX:014539, RRID:IMSR_JAX:014539) were purchased from Jackson Laboratory. cfos-tTA (cfos mice) were derived from the Mutant Mouse Regional Resource Center (stock no. 031756-MU, RRID:MMRRC_031756-MU). All transgenic mice were backcrossed in a C57BL6/J background more than 10 times. All transgenes were kept as heterozygotes in the chromosome used for the experiments to avoid possible complications of overexpressing Cre recombinase, tTA, and loss of the Rosa allele. Both male and female F1 mice were used (the exact number of each sex, along with mouse ID numbers, is provided in Mendeley Data). Mice were group-housed 2-5 per cage before surgery. Age-matched littermates were used for comparisons with nestin/cfos/jaws and nestin/jaws mice (or cfos/jaws and jaws mice in Fig. 3g-h) to avoid potential confounds of litter, age, cage, Tamoxifen (TMX) and Doxycycline (Dox) administration, or light delivery. Nestin, cfos, GCaMP6s, jaws, and halo genotypes were confirmed by tail genotyping. Primers used were as follows. For nestin mice, Cre-forward: CATCTGCCACCAGCCAGCTATCAACTCG, ERT2-reverse: ACTGAAGGGTCTGGTAGGATCATACTCG; successful PCR amplification produced a 430-bp band. For GCaMP6s mice, GCaMP6s-forward: TGGGGATGGTCAGGTAAACT, GCaMP6s-reverse: CCACATAGCGTAAAAGGAGCA; successful PCR amplification produced a 165-bp band. For Jaws mice, Jaws-forward: ACAGTG TCTGGGAGTGGAATG, Jaws-reverse: TGGTGCAGATGAACTTCAGG; successful PCR amplification produced a 220-bp band. For cfos mice, cfos-forward: TGCTCCCATTCATCAGTTCC, cfos-reverse: ACCTGGA-CATGCTGTGATAA; successful PCR amplification produced a 326-bp band. For Halo mice, Halo-forward: TGGATGTTCCATCTGCTTCTG, YFP-reverse: TTGCCGGTGGTGCAGATGAA; successful PCR amplification produced a 700-bp band.

### Drugs

**Tamoxifen injection.** To create *TRE-GCaMP6s*, *TRE-Jaws-GFP*, or *pCAG-eNpH3.0-YFP* transgenes in ABNs, all F1 mice were treated with TMX at 6-8 weeks of age to ensure that target neurons were from the same neural/stem progenitor pool and that phenotypic differences between mice were not attributed to nonspecific TMX effects. TMX (120 mg/kg) was injected into the peritoneal cavity five times at 1- or 2-day intervals, with completion of the injection period within 10 days. The period between the central date of TMX injection and fear conditioning was kept at 4 weeks to make the targeted ABN age consistent across experiments, except for the experiment in which it was 10 weeks to tag 10-week-old ABNs (Fig. 3e-f). To prepare TMX solution, 30 mg TMX (T5648, Merck, USA) was dissolved in 100 ml of 100% EtOH and added to 1000 ml sunflower oil (Wako, Japan), and the solution was placed in an ultrasonic cleaner for further dissolution. As a final step, EtOH was evaporated by a centrifuge. We did not make a comparison between TMX-injected and non-injected mice to avoid potential confounding effects of TMX injections.

**Dox treatment.** To stop cfos-tTA-driven transgene expression (i.e., Jaws-GFP) in ABNs, Dox (D9891, Sigma-Aldrich, USA) was added to mouse food at 40 mg/kg two-weeks after the TMX injection, except for Fig. 2c Group 1 (three-weeks after the TMX injection) and the No Dox groups. For targeting GNs (i.e., cfos/jaws and jaws mice with AAV CaMKII-Cre injection in Fig. 3g-h), we did not inject TMX but instead matched the mouse age to provide Dox according to the protocol targeting ABNs (as Fig. 2c Group 3 without TMX injection).

### Histology

Four weeks after TMX injection, mice were perfused transcardially with 0.1 M PBS followed by 4% paraformaldehyde (PFA). Brains were removed, fixed overnight in PFA, and transferred to PBS. Coronal sections (50 μm) were cut using a vibratome (VT1200S, Leica). For counting GFP+ cells, sections were incubated overnight with rabbit polyclonal anti-GFP primary antibodies (1:250, Thermo Fisher Scientific Cat# A-11122, RRID:AB_221569), then for 60 min with anti-rabbit HRP secondary antibodies (1:500, Jackson ImmunoResearch Labs Cat# 711-036-152, RRID:AB_2340590) at room temperature. Signal amplification was achieved using tyramide signal amplification (biotinylated-TSA, original)[9], followed by incubation for 30 min with streptavidin-Alexa-488 (1:500, ImmunoResearch Labs Cat# 016-540-084, RRID:AB_2337249) at room temperature. Sections were mounted on slides with a mounting medium (Permafluor, Thermo Fisher Scientific Cat# TA-030-FA) containing DAPI (Merck). GFP+ cells in the bilateral suprapyramidal blades of the dentate gyrus (DG) were counted in the anterior-posterior direction from AP −1.3 to −2.7 mm relative to bregma, covering the effective area of light stimulation (i.e., the tip of the optic cannula was placed at the sulcus lacunosum-moleculare at the position of AP −2.0 mm)[9]. Imaging and quantification of GFP+ cells were performed manually using a microscope (Imager M2, Zeiss) with 10× and 20× objectives. Density is calculated by dividing the absolute number of GFP+ cells by the volume of the granular cell layer[9].

### Blinding procedure

To minimise cognitive bias and expectation effects, we ensured that the individuals responsible for freezing and statistical analyses were unaware of the identity of the mice. This measure helped maintain the objectivity of the analyses and minimised potential bias. In the experiment described in Supplementary Fig. 5a-b, we implemented a more stringent blinding procedure. Separate experimenters were assigned to each step of the process, including genotyping, assigning and scheduling, surgery, fear conditioning, optogenetics, scoring freezing, and performing statistical analysis. The person responsible for genotyping was aware of the mice's identities but was not informed of the experiment's objectives and did not participate in the experimental design or further experiments. Each experimenter involved in subsequent steps was kept unaware of the identity of the mice during all procedures and analyses. Mouse identification codes were assigned and managed by a separate individual who did not participate in the experiments, ensuring that the experimenters' actions and interpretations remained objective and unbiased.

### Surgery

**Implantation of lens and electroencephalogram (EEG)/electromyography (EMG) electrodes.** At 11 weeks of age, mice were anaesthetized with isoflurane and fixed in a stereotaxic frame (Stoelting, USA). The height of bregma and lambda were adjusted to be level. The microendoscope lens (1-mm diameter, 4-mm length, Inscopix, USA) was placed at AP −2.0 mm, ML + 1.2 mm, and DV −1.95 mm relative to bregma. After implanting the lens, EEG electrodes consisting of stainless-steel recording screws were implanted epidurally at AP + 1.5 and −3 mm and ML −1.7 mm. One week after electrode placement, the baseplate for a miniaturized microendoscope camera (nVoke, Inscopix, USA) was attached above the implanted microendoscope lens. After

baseplate surgery, mice were habituated to the cable for 6 days and the attached microendoscope camera for 3 days before recording.

**EEG/EMG electrode and optic fiber implantation.** At 9–11 weeks of age (or 16 ~ 17 weeks of age for tagging 10-week-old ABNs), mice were anesthetized with isoflurane and fixed in a stereotaxic frame. The height of bregma and lambda were adjusted to be level. EEG electrodes were implanted epidurally at AP + 1.5 and −3.0 mm and ML + 1.5 and −1.7 mm, and EMG electrodes consisting of stainless steel Teflon-coated wires (AS633, Cooner Wire, USA) were placed bilaterally into the trapezius muscles. Optic cannulas (200-mm diameter, 5-mm length optic fiber plus zirconia connector, Thorlabs, Japan) were placed at AP −2.00 mm, ML ±1.10 mm, and DV −1.95 mm (Supplementary Fig. 4d).

**Optic cannula with local field potential (LFP) electrode implantation.** For DG LFP recordings, an unsheathed portion of platinum wire was attached to the tip of the optic fiber. Mice were anesthetized with isoflurane and fixed in a stereotaxic frame. The height of bregma and lambda were adjusted to be level. At 10 weeks of age, EEG electrodes were placed at AP + 1.5 and −3.0 mm and ML + 1.5 and +1.7 mm relative to bregma. For the reference electrode, an EEG electrode was placed at AP −5.0 mm and ML 0.0 mm relative to bregma. An optic cannula attached to an LFP electrode was placed at AP + 2.0 mm, ML ± 1.35 mm, and DV −1.78 mm relative to bregma (Supplementary Fig. 6a). Mice were habituated to being chronically tethered in the recording chamber for 7-8 days until they exhibited a regular sleep-wake cycle. After completion of each experiment, randomly chosen mice were subject to histological analysis to confirm optic cannula location.

### Granule neuron tagging and manipulation
**Virus preparation.** Adeno-associated virus (AAV) vector was prepared as previously described[46]. Briefly, AAV was generated by tripartite $Ca^{2+}$ phosphate transfection (AAV-1 expression plasmid, Penn Vector Core), adenovirus helper plasmid (Agilent), and pAAV plasmid pENN. AAV.-CamKII 0.4.Cre.SV40 (Addgene plasmid #105558, RRID:Addgene_105558) into 293 A cells. After 3 days in 5% $CO_2$ at 37 °C, 293 A cells were re-suspended in artificial cerebrospinal fluid, frozen and thawed four times, and treated with benzonase nuclease (Millipore Sigma, Cat #E1014) to degrade all forms of DNA and RNA. Cell debris were removed by centrifugation, and the virus titer in the supernatant was determined using real-time polymerase chain reaction ( $> 1 \times 10^{13}$ viral genome/ml).

**Virus injection.** We used the same volume (70 nl) of AAV:CaMKII-Cre vector from only one preparation (i.e., single batch) and the same duration between virus injection and optogenetic manipulation (26 days) across all AAV experiments. Adult mice (9-10-week-old cfos/jaws, cfos/Ai94, or jaws mice) were anesthetized with isoflurane and fixed in a stereotaxic frame (Stoelting, USA). AAV solution was injected into the dorsal hippocampus at AP + 2.0 mm, ML ± 1.2 mm, and DV 1.6 mm relative to bregma. The virus was injected using a microinjector air pressure-based injection system (IM 300, Narishige International USA, Inc.) connected to a glass pipet injector. Injections were performed for >15 min. After microinjection, the injector needle was left in place for 5 min and then slowly withdrawn. Mice were allowed to recover for 2 weeks before optic cannula or baseplate implantation for the silencing/imaging experiments.

### Behaviour
Mice were habituated to experimenter handling by two or three 2-min handling sessions per day for a total of 11 sessions before behavioural experiments. A shock reactivity index to the first shock delivered during fear conditioning was calculated based on the velocity (V) of mouse movement during the 2 s immediately before and during the shock as follows:

$$(V_{during\ shock} - V_{pre-shock})/(V_{during\ shock} + V_{pre-shock}) \qquad (1)$$

Freezing behaviour was measured using an automated scoring system (Freezeframe, Med Associates, USA) and was defined as a ≥ 0.25 s continuous absence of movements except for breathing.

**Context exposure.** Context A consisted of a stainless steel conditioning chamber (width × depth × height, 31 × 24 × 21 cm) containing a stainless steel grid floor. The grid floor was composed of bars (3.2-mm diameter) spaced 7.9 mm apart that allowed the delivery of electric shocks. Under the grid floor was a stainless steel drop pan lightly cleaned with 75% EtOH, which provided a background odor. The front, top, and back of the chamber were made of clear acrylic, and the two sides were made of aluminum panels. A camera was placed at the back of the chamber, and the front of the chamber was covered by an exterior black curtain. Mice (in Figs. 1, 2c-d, 2i-l, 3a-b, 3e-h and Supplementary Figs. 2, 3a-c, 4, 5a-b, 5k-l) were exposed to context A twice per day for 3 days (10 min each, at least 3 h apart within a day) under Dox-off. Mice in Fig. 2i-l were connected to an optic cable during Test 1 and 2.

Context C consisted of a cylindrical clear Plexiglas cage (21.9-cm diameter, 31.6-cm height) with a floor that was lightly cleaned with water. A camera was placed at the top of the chamber, and the chamber sides were covered with aluminum foil. Mice in Supplementary Fig. 5d-e were exposed to context A as described above but under Dox-ON, remained in their home cage for 1 day under Dox-On, and were then exposed to context C twice per day for 3 days (10 min each, at least 3 h apart within a day) under Dox-Off. Mice in Fig. 3e-f underwent a procedure similar to the mice in Supplementary Fig. 5d-e, except they were not exposed to context A before context C.

**Contextual fear conditioning.** In the section described as "Conditioning" in the figures, mice were placed in the conditioning context (i.e., context A) for a total of 600 s, with 2-s foot shocks (0.75 mA) administered at 298, 388, 478, and 568 s. The training was conducted between 10:30 and 11:30 am (zeitgeber time (ZT) 1.5-2.5). To assess contextual fear memory, mice were returned to the conditioned context for 5 min without the delivery of foot shocks (i.e., retrieval test) the next day. The mice in Fig. 2i-l were tested twice with or without 589-nm light stimulation on day 1 and 2 after conditioning, respectively.

**Trace fear conditioning.** Mice in Fig. 4 were placed in context A, and 180 s after habituation to the context, an auditory stimulus (1-Hz sweep, 85 dB) was presented for 20 s, followed by an 18-s trace interval and 2-s foot shock (1.0 mA). This sequence (220-s) was repeated five times, totaling 18 min of training. The training was conducted between 10:30 and 11:30 am (zeitgeber time (ZT) 1.5-2.5). Twenty-four hours later, mice were returned to Context A for 5 min without the delivery of foot shocks (i.e., context test). Three hours after the context test, mice were placed in context B for 6 min, and the tone was played during the last 180 s (i.e., tone test). Context B was similar to the conditioning context except that the floor and sides of the chamber were covered with white plastic boards, a piece of cardboard with a blue and white rectangular pattern was affixed to the back wall, and water instead of ethanol was used for the background odor.

**LFP/EEG/EMG recording and sleep state analysis.** As previously described[9], LFP/EEG/EMG signals were recorded in the home cage after conditioning using a data acquisition system (Vital Recorder or SleepSign Recorder, KISSEI COMTEC, Japan). Briefly, EEG/EMG data were collected at a sampling rate of 128 or 512 Hz. Coaxial electric and optic (Doric Lenses, Canada) slip rings allowed mice to move during recording. Real-time sleep state analysis and optogenetic intervention

was conducted either automatically by the artificial intelligence (UTSN-L2) equipped system[47] or manually by experimenters. Manual analysis was based on visual characteristics of EEG and EMG waveforms with consistent timing indicators on the display for wave frequency determination and video surveillance of mouse movement. Wakefulness was defined as continuous mouse movement or de-synchronised low-amplitude EEG with tonic EMG activity. Non-REM (NREM) sleep was defined as dominant high-amplitude, low-frequency delta waves (1–4 Hz) accompanied by less EMG activity than that observed during wakefulness. REM sleep was defined as dominant theta rhythm (6–9 Hz) and the absence of tonic muscle activity. If a 10-s epoch contained more than one sleep state (i.e., wakefulness, NREM sleep, or REM sleep), the predominant state was assigned to the epoch. To avoid any confounding effects, offline sleep architecture analysis was performed by one individual who was completely blind to the objectives of the analysis and the identity of mice.

## Imaging

**Imaging ABN/GN activity**. Recording of GCaMP signals from the DG of freely behaving mice was performed over 4 days using a microendoscope. During the first 3 days, recordings were performed when mice were exposed to context A twice a day (a total of 6 times, with each session 10 min and more than 3 h apart between sessions). On day 4, recording was performed for 10 min in context A before shock delivery in context A (i.e., preS). We then detached the microendoscope (< 1 min) to avoid a change in the field of view due to hitting the microendoscope against the wall during the shock. Then, the conditioning protocol was performed: mice were placed in context A for a total of 600 s, with 2-s foot shocks (0.75 mA) at 298, 388, 478, and 568 s. Immediately afterward, we re-attached the microendoscope (< 1 min) and performed 10 min of recording after the shock in context A (i.e., postS). The conditioning was conducted between 10:30 and 11:30 am (zeitgeber time (ZT) 1.5-2.5). After conditioning, the microendoscope remained attached to the mouse's head for the following 2.5-hour period. During this period, we performed recording when the mice were in REM sleep to minimise potential photo-bleaching caused by continuous light exposure to GCaMP.

6 h after the conditioning, the mice were returned to the conditioned context with for 10 min with the microendoscope.

**Ca²⁺ imaging data processing**. Ca$^{2+}$ signals were extracted and tracked across sessions utilizing CaliAli release v1-beta[13]. CaliAli is specifically designed for extracting and tracking neuron activities across multi-session imaging experiments, ensuring consistency in the number of neurons tracked across all sessions. Initially, raw Ca$^{2+}$ imaging videos underwent spatial downsampling by a factor of 2 and were then motion-corrected using blood vessels and log-demon image registration. Inter-session misalignments were corrected, and sessions were concatenated using CaliAli. Ca$^{2+}$ signals were extracted from the concatenated video using a spatial filter size ('gSig') set to 2.5. The minimum peak-to-noise ratio (PNR) and correlation threshold for seed initialization were determined individually for each recording based on careful monitoring of the PNR and correlation images. Automatic merging and removal of false positives were performed using default thresholds as previously defined[48]. Sequential updates of temporal, spatial, and background components, and automatic merging and deletion of components continued until changes across iterations were <5% in terms of cosine similarity. Ca$^{2+}$ trace deconvolution was performed using the FOOPSI algorithm with a first order autoregressive (AR) model during initialization and CNMF. The final Ca$^{2+}$ transients were noise-scaled and deconvolved using the thresholded-FOOPSI algorithm and second order AR model, which is slower but more accurate. Finally, false-positive extractions with temporal and spatial characteristics not corresponding to putative somatic Ca$^{2+}$ signals were discarded based on visual inspection of the detected components.

## Optogenetics

**Patch clamping**. For electrophysiological confirmation of silencing[9] using Jaws, we injected mice with pentylenetetrazole (PTZ) to overcome the sparseness in ABN tagging. Without this procedure, it was practically impossible to patch viable ABNs in slice preparation. Eleven-week-old nestin/cfos/jaws mice received one to five injections of PTZ (2 mg PTZ in 1 ml PBS; 10 μg/g) at 10-min intervals until a partial seizure was induced, after which coronal brain slices were prepared. Current injection was performed with a patch-clamp electrode at the soma. Orange light (589-nm laser; Shanghai Lasers, China) was used to stimulate Jaws. Light was focused into the back focal plane of a LUM-PLAN 40× lens and exited as a parallel beam. The diameter and total area of the illuminated area were 1 mm and 0.78 mm², respectively. The light power density was 66 mW/cm².

**Optogenetic silencing**. Orange light (589-nm laser, 20 mW at the tip of the optic cannula, bilateral) was used to stimulate Jaws in the dorsal DG in freely moving mice. For light delivery to nestin/cfos/jaws mice during REM sleep, a generator producing 0.05-Hz transistor-transistor logic pulses (10-s on/off cycles) was used to control the lasers as previously reported[9]. The average total duration of silencing during REM sleep was 13.6 min. For light delivery during contextual fear memory tests, orange light was pulsed at 30 Hz.

For nestin/halo mice, a single pulse of orange light (25 ms per pulse, 589 nm, 20 mW) was delivered unilaterally to the ABNs at the target phase of each local theta oscillation cycle during REM sleep. A customized closed-loop feedback device (CLFD) controlled the timing of light stimulation at the target theta phase (based on STM32, STMicroelectronics, Switzerland). CLFD sampled the ipsilateral local field potential (LFP) and detected the timing of the peak or trough of each theta cycle (code available in Mendeley Data). It then controlled light delivery at the target phase within the next theta cycle based on the preset parameters (e.g., target phase).

For targeting Phase 3 (trough phase) and Phase 4 (descending phase), CLFD first detected a data point (of LFP amplitude) higher than a pre-set threshold value, which was determined based on the standard deviation of the LFP signal during theta oscillation. It then compared the next two data points. If the latter data point value was lower than the earlier value, it regarded the timing of the earlier data point as the peak position of the cycle. For targeting Phase 1 (ascending phase) and Phase 2 (peak phase), CLFD detects the trough in a similar manner, but in the opposite direction to estimate the target timing.

We used a yoked control design to specifically examine the effects of silencing ABNs during particular phases of theta oscillation. In our setup, we connected two mice: the experimental mouse, in which ABN activity was silenced precisely during a single designated theta phase using CLFD, and the yoked control mouse, which received light pulses at the same times but not synchronised to any specific neural activity, brain states (e.g., wakefulness or sleep), or behaviour. Not all experimental mice had corresponding yoked controls; therefore, the total number of yoked control mice does not equal the total number of experimental mice. The assignment of mice to experimental groups targeting specific theta phases was done randomly, regardless of whether they had a yoked control. This approach ensures that any observed changes in the experimental mice are due to the targeted silencing of ABNs at a specific theta phase rather than nonspecific effects of light exposure.

The sensitivity and precision of light delivery were calculated by comparing online and offline judgements of sleep states. Briefly, we first subdivided the time window of light delivery into 10-s epochs offline and classified each epoch into true positive (TP: light was correctly on), false positive (FP: incorrectly on), false negative (FN: incorrectly off), or true negative (TN: correctly off) categories. Sensitivity was calculated as TP/(TP + FN), and precision was calculated as TP/(TP + FP).

For light delivery at the targeted theta phase during REM sleep in Fig. 4, data from six REM episodes per mouse, including LFP and laser trigger signals, were extracted. The LFP signal was filtered using a 6-9 Hz band-pass filter and then transformed into phase angles using the Hilbert transform. The centers of the 25-ms trigger signals were aligned with the corresponding LFP angles.

## Statistical analysis

Statistical analysis was performed using GraphPad Prism version 9 (GraphPad Software, USA), custom scripts in MATLAB (MathWorks, USA) and R. Type I error was set at 0.05. Other details of statistical analyses are described in the figure legends and Supplementary Table 3. In all figures, error bars indicate SEM unless stated; ns, not significant; $*p < 0.05$, $**p < 0.01$, $***p < 0.001$.

**Average Ca$^{2+}$ activity analysis.** Analysis was done based on the predicted spikes amplitudes of the raw Ca$^{2+}$ transients (S, in CNMF-E notation). To compare the mean activities of ABNs and GNs, each neuron's activity was analysed as a percentage of its total activity across all sessions using the following mixed linear model (in Wilkinson notation):

$$Activity = 1 + Time \cdot NeuronCategory + Shock \cdot NeuronCategory + REM \cdot NeuronCategory \qquad (2)$$

In this model, Time was a continuous variable reflecting recording time relative to the first session, Shock was a categorical variable indicating session occurrence before or after shock experience, and REM was a categorical variable indicating recording during REM sleep. NeuronCategory indicates the ABN vs. GN categorization of neurons.

To account for potential variations across mice, random effects were included to model differences in intercepts and slopes among mice. The statistical significance of the fitted parameters was assessed using an F-test. Given the non-normally distributed residuals, we used random permutation to confirm the significance of our results. This involved shuffling the residuals randomly while maintaining the model structure. By re-evaluating the model with these permuted residuals, we could assess whether the observed F-values were statistically significant beyond random chance (95% confidence).

**Ca$^{2+}$ activity PV correlations analysis.** For each recorded session, PVs were constructed using the average spike amplitude of neurons as components of the vector. Correlation matrices were then created by calculating the correlation between PV obtained across each session pair. The PV correlations were modeled utilizing the same linear model used to describe neuron activities applied to the PV correlation data. In this model, the predictors include the pairwise Euclidean distances between sessions based on Time, Shock, and REM/Retrieval status, along with the categorical variable *NeuronCategory*. The statistical significance of the fitted parameters was assessed using an F-test.

This analysis was also repeated under three additional conditions: (1) by including the number of neurons as a covariate in the model; (2) by re-computing PVs after randomly downsampling activity to equalize the duration of wake and REM periods (1000 replicates); and (3) by randomly downsampling the number of GNs to match the size of the ABN population (1000 replicates).

**Modeling Freezing behaviours as function of light delivery in different phases of theta.** The effect size of freezing depending on the timing and the amount of silencing was estimated using 10-degree-binned histograms of the theta-phase light delivery for each mouse (Supplementary Fig. 8). To address the correlation between nearby phases and the larger number of predictors compared to observations, we employed Ridge regression. This method introduces a penalty term ($\lambda$) into the linear regression error function to penalize large coefficients ($\beta$) and mitigate multicollinearity and overfitting. The optimal $\lambda$ was determined by minimising a 5-fold cross-validation error. The obtained $\beta$ coefficients reflect the relative changes in freezing behaviour resulting from light stimulation in the corresponding phase of REM theta. In real experiments (i.e., Fig. 4), perfectly stimulating at a specific phase of theta is unfeasible. Therefore, to estimate the predicted freezing obtained by targeting specific phases of theta, we multiplied the $\beta$ coefficients by the stimulation density and variability observed in real experiments. These patterns were derived by centering and averaging the theta-phase light delivery histogram of each mouse, systematically shifted to align with different theta phases. This approach enabled us to predict freezing behaviours for different targeted REM theta phases while considering realistic stimulation patterns.

## Reporting summary

Further information on research design is available in the Nature Portfolio Reporting Summary linked to this article.

## Data availability

No unique reagents were generated in this study. Data for LFP and phase-specific silencing study is available at Mendeley Data [https://doi.org/10.17632/2xfktwhj8x.1]. All detailed statistics are included in the Supplementary information. All other data supporting this study are available from the corresponding authors. Source data are provided with this paper.

## Code availability

The datasets and custom code used to perform the non-trivial quantitative analyses are available on GitHub [https://github.com/vergaloy/Srinivasan_Koyanagi_2025]. A preserved snapshot of the code at the time of publication has also been archived on Zenodo [https://zenodo.org/records/15803872].

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

## Acknowledgements

We thank K.G. Akers for comments on the manuscript, Y. Mimura, R. Shiina, E. Watanabe, P. Wu, and D. Kumar for technical assistance, and M. Sakurai, I. Sekiguchi, and M. Yoshida for secretarial supports. This work was partially supported by Japan Agency for Medical Research and Development (JP21zf0127005, JP23wm0525003), Japan Society for the Promotion of Science (JSPS) (24H00894, 23H02784, 22H00469, JP16H06280, 20H03552, 21H05674, 21F21080), Takeda Science Foundation, Uehara Memorial Foundation, and The Mitsubishi Foundation to MS; JSPS(19F19310) to M.S. and S.S.; JSPS(21F21080) to M.S. and P.V.; Goho Life Sciences International Fund, The Tokyo Biochemical Research Foundation Postdoctoral Fellowships for Asian Researcher in Japan, Takeda Science Foundation Scholarship for Foreign Researchers to SS; JSPS(21J11746, 24K18212, 23K19393 and DC2 fellowship) to I.K.

## Author contributions

S.S. and I.K. contributed equally to this work. Conceptualization: M.S., S.S., and I.K.; Methodology: S.S., I.K., T.T., K.V., Y.C., P.V., and M.S.; Investigation: S.S., I.K., P.V., Y.W., A.O., K.V., Y.C., T.N., and N.K.; Validation: I.K., T.N., and M.S.; Formal analysis: P.V., I.K., and S.S.; Data curation: S.S., I.K., P.V., and M.S.; Visualization: P.V., S.S., I.K., K.V., and M.S.; Writing – original draft: S.S., I.K., P.V., and M.S.; Writing – review &

editing: M.S., S.S., I.K., and P.V. Funding acquisition: M.S., S.S., and I.K. Resources: M.S. and T.S.; Supervision: M.S. and T.S.; Project administration: M.S.

## Competing interests

The authors declare no competing interests.
