## [Transparent Peer Review file · Nature Communications]

Transient reactivation of small ensembles of adult-born neurons during REM sleep supports memory consolidation in mice

Corresponding Author: Dr Masanori Sakaguchi

Version 0:

Reviewer comments:

Reviewer #1

(Remarks to the Author)

The authors have previously shown that ABNs are required for post-learning memory consolidation during sleep. This manuscript shows that adult born neurons (ABNs) have more stable representations of experience than developmentally born neurons (DBNs), in terms of population vectors, over time and from waking experience to REM sleep. The authors then specifically tag a small population of adult born engram neurons (only 5 neurons/mouse) and silence those neurons during post-learning sleep and during memory retrieval. Remarkably, they find that their activity during sleep is necessary for subsequent memory recall. This effect is specific to immature ABNs because there is no effect when older ABNs or developmentally born neurons are silenced. Finally, the authors silence these neurons during specific phases of theta (during REM sleep) and find their greatest memory deficits when ABN engram neurons are silenced during a specific phase of theta. Overall the manuscript is well written and the study is well conducted. More importantly, it is very interesting (and surprising) that silencing as few as 5 neurons is sufficient to disrupt memory recall, provided that it occurs during sleep. I have a few comments and suggestions for improvement.

I don't fully understand the rationale for using the nestin/cfos/G6s mouse to image ABNs. The authors claim it is to allow them to only visualize activity in differentiated ABNs. This makes sense, since only functional ABNs will express fos. But it also means that any ABN that had an active fos promoter over the next 4w is going to express Gcamp, right? This is sort of an unusual or biased selection of ABNs possibly. It will only tag neurons that had been active at some point but not other perhaps functional neurons that have not yet expressed fos (if such a thing exists). This isn't necessarily an issue, and I'm not sure there's a better way to accomplish what the authors are trying to do, but it is worth thinking about and maybe even mentioning in the discussion briefly.

For the experiments in Fig 2, when was TMX injected relative to Dox-On in the home cage? It looks like it was injected at day -14 but then the tagged ABNs would only be ~2 weeks old at the time of conditioning, and 3w old at the time of retrieval testing. This is too young to have formed synapses and make any behavioral contribution. In Fig 3 I can see that TMX was given at -4w or -10w. Can that be made more clear in Fig 2? If TMX was given at -4w (for example) in Fig 2, was Dox on or off?

I don't fully appreciate the significance of the phase-specific silencing effect. Can anything more specific be stated or speculated as to why silencing at the peak has strongest effects, or is more relevant or important?

How confident are the authors that the neurons expressing Jaws are really adult born? The expression in the supplementary data shows cells in the middle or superficial granule cell layer, but this is not where ABNs reside. The neurons in Fig 2 are somewhat more in the expected location. Do these cells have a high input resistance as expected? How consistently did the authors find these cells in the inner granule cell layer, or near the sgz?

Minor: last sentence of abstract: "Overall, this study provides mechanistic insights into how new neurons integrate into functional circuitry in the adult brain." Does it really? It seems to focus more on the functional role of new neurons during sleep. Perhaps the authors can adjust or reword. Same for the last sentence of the discussion. It is very dramatic and seems off topic. Maybe the authors can be a little more specific with respect to how ABNs can be "therapeutically applied in cases

of neuronal loss”.

Lines 92-93: can you really say that jaws was expressed in ABNs “that were active during context exposures”? I think all you can claim with certainty is that jaws is expressed in ABNs (unless Ca²⁺ activity was visualized in these neurons, for example)

(Remarks on code availability)

Reviewer #2

(Remarks to the Author)

This article is a follow up on findings presented in a Neuron paper published in 2020. That paper demonstrated that a small subset of adult born neurons in DG are active during post-fear conditioning REM, and that that population's activity during post-conditioning REM benefits memory consolidation. This article makes an advance upon that finding, by focusing only on the subpopulation of neurons that are active during learning. The result is as would be expected, in the affirmative - disrupting activity in this subpopulation similarly disrupts consolidation. While the techniques employed are interesting, and the original finding reported in 2020 is also interesting, the conclusions are generally the same - i.e. this doesn't really change the interpretation of the data from the 2020 paper. Aside from this general concern about the degree of advance, there are a few additional concerns, which are listed below:

- 1) The learning paradigm employed has not been shown to produce a form of memory that requires sleep for consolidation - much less REM sleep. The importance of the findings would be more clear if it was clear that REM sleep was vital to consolidating this memory in the first place.
- 2) There is a concern about the conclusion on Page 4: "These results suggest that ABN ensemble activity represents the original memory more faithfully than GN ensemble activity during REM sleep within the memory consolidation period." Can this really be fairly concluded? The number of ABNs means that they form a tiny ensemble, while GNs are a significantly larger one (by orders of magnitude). So doesn't it stand to reason that with orders of magnitude more degrees of freedom for the GNs, there would almost by definition be more variability in their activity patterns across time? It isn't clear that this comparison is statistically appropriate.
- 3) Related to the analysis in the point above, the conclusion about the effect sizes in REM, in particular, may be statistically spurious, given the smaller number of time points for comparison in REM vs. other states. To make this more problematic, REM is known to be suppressed after fear conditioning in mice, meaning that the number of comparisons is selectively reduced relative to other states.
- 4) From page 5: "Consistent with previous findings, only a few ABNs were tagged in cfos+ (Fig. 2b-d), with a mean (standard deviation) of 4.7(4.7) neurons per mouse in the light-effective area". While it is clear that this population has to be small, almost by definition, how are readers to feel confident that this population even exists across all mice? Isn't it plausible that, given the SD, the cfos+ ABN population in some mice will actually be zero neurons?
- 5) A very critical piece of information for interpreting all of these data is confirmation that (e.g.) Jaws-GFP expression is restricted to DG, vs present in other brain areas. Given the cropping of the images, this is an unanswered question - this it is impossible to feel confident in the behavioral results.

(Remarks on code availability)

Reviewer #3

(Remarks to the Author)

The tour de force experiments described in the manuscript, “Adult-born neuron reactivation in REM sleep for memory consolidation” by Srinivasan et al., give invaluable insights on the role of adult neurogenesis in memory and, in so doing, show fundamental mechanisms underlying sleep-dependent consolidation. Using incredibly difficult genetic tracing and optogenetic silencing to establish the necessity of ABNs during REM sleep, the authors convincingly demonstrate the presence of ABN memory ensembles (cfos+ engrams) specific to memory consolidation and distinguishing them uniquely from unrelated ensembles (cfos- engrams) using contextual fear conditioning and trace fear conditioning. It is especially surprising that just 5 ABN may be sufficient to establish the necessary engram. My main concern is related to Fig 4 where the authors propose that these ABN memory consolidation ensembles can be further defined by being phase-locked to theta cycles during REM (see below). By addressing my concerns, I believe that this manuscript will be well-suited to Nature Communications given its significant technical and conceptual advances which will be of great interest to a wide audience, especially those interested in neurogenesis, engrams, and sleep.

Major concerns:

- 1) It is unclear why the authors used a version of a contextual fear conditioning task involving exposure to Context A six times over three days. Based on the references they cited, the authors are aware of the studies by Denny et al (Hippocampus, 2012) and especially Drew et al (Behav Neurosci, 2010) showing that this much pre-exposure is not

necessary and may in fact complicate interpretation of results. The authors should provide adequate justification for the use of this revised contextual fear conditioning protocol.

2) Fig. 4 – the authors need to clarify why was trace fear conditioning included in their studies. It seems unnecessary given their strong use of contextual fear conditioning in previous figures. Including trace fear conditioning puts into question whether their phase-locking results are only possible with this behavior and not the contextual fear conditioning paradigm. The authors therefore will need to provide adequate justification or provide additional experiments with contextual fear conditioning only and contrast results with trace fear conditioning.

3) Fig. 4 – the authors compared Phase 1, 2, 3, and 4 only to the yoke controls. They need to show statistical analysis between phases in order to make the argument that ABN memory consolidation ensembles are locked to a specific phase of the theta cycle. For instance, in Fig. 4e (context test), it appears that Phase 1 may be significantly different than Phase 3 and 4 but possibly not Phase 2. However, in Fig 4e (trace test), it appears that there is no significant difference between phases. If true, then together these results may even strengthen the authors' argument. If not, then the authors will have to more accurately qualify their conclusions on phase-locking.

Minor concerns:

1) Line 86 – the authors indicate that ABN ensemble activity more faithfully represents the original memory than GN. The authors need to clarify what they mean by “more faithfully”. Could the more stable PV they calculated for ABN possibly be due to their higher baseline activity levels than GN, and does not necessarily mean that memories are more faithfully encoded in ABNs over GNs?

2) Line 119 – the authors state that there is low overlap between conditioning and retrieval ABN populations and cite their previous work. It would be helpful to readers not aware of their past studies to show actual results based of their current experiments, even if in the Supplement.

3) Line 205-206 – the authors claim that they provide new insight into how new neurons can be therapeutically applied in cases of devastating neuronal loss. I am not so sure how their studies show this. It would be best to delete this part.

4) The title of the manuscript is a little wonky and does not appear to encapsulate the exciting results. May I suggest changing it to something like: “Reactivation of adult-born neuronal ensembles is necessary for memory consolidation during REM sleep”. Of course, this is merely a suggestion and does not affect my enthusiasm for the manuscript.

(Remarks on code availability)

Version 1:

Reviewer comments:

Reviewer #1

(Remarks to the Author)

The authors have worked hard to sufficiently address all of my concerns. Accept.

(Remarks on code availability)

Reviewer #2

(Remarks to the Author)

The authors have done a stellar job of working to respond to prior concerns brought up during review. With one exception, these concerns have been addressed. One final issue should be resolved prior to acceptance of the manuscript, in order to provide transparency to readers.

This outstanding issue, raised in the prior review, is referred to in the authors' response letter:

Comment 5

“5) A very critical piece of information for interpreting all of these data is confirmation that (e.g.) Jaws-GFP expression is restricted to DG, vs present in other brain areas. Given the cropping of the images, this is an unanswered question - this it is impossible to feel confident in the behavioral results.”

The authors claim to have included a zoomed-out view, but this is not evident in the revised files. Perhaps the original comment was unclear, but what should be presented is a VERY zoomed out view, providing imaging of not only DG, but areas outside DG - i.e., full hippocampal sections, or better yet, full coronal brain sections.

(Remarks on code availability)

Reviewer #3

(Remarks to the Author)

The authors have thoroughly addressed all my concerns. I strongly endorse the acceptance of this manuscript.

(Remarks on code availability)

Reviewer 1

Comment 1

“The authors have previously shown that ABNs are required for post-learning memory consolidation during sleep. This manuscript shows that adult born neurons (ABNs) have more stable representations of experience than developmentally born neurons (DBNs), in terms of population vectors, over time and from waking experience to REM sleep. The authors then specifically tag a small population of adult born engram neurons (only 5 neurons/mouse) and silence those neurons during post-learning sleep and during memory retrieval. Remarkably, they find that their activity during sleep is necessary for subsequent memory recall. This effect is specific to immature ABNs because there is no effect when older ABNs or developmentally born neurons are silenced. Finally, the authors silence these neurons during specific phases of theta (during REM sleep) and find their greatest memory deficits when ABN engram neurons are silenced during a specific phase of theta. Overall the manuscript is well written and the study is well conducted. More importantly, it is very interesting (and surprising) that silencing as few as 5 neurons is sufficient to disrupt memory recall, provided that it occurs during sleep. I have a few comments and suggestions for improvement.

I do not fully understand the rationale for using the nestin/cfos/G6s mouse to image ABNs. The authors claim it is to allow them to only visualize activity in differentiated ABNs. This makes sense, since only functional ABNs will express fos. But it also means that any ABN that had an active fos promoter over the next 4w is going to express Gcamp, right? This is sort of an unusual or biased selection of ABNs possibly. It will only tag neurons that had been active at some point but not other perhaps functional neurons that have not yet expressed fos (if such a thing exists). This isn't necessarily an issue, and I'm not sure there's a better way to accomplish what the authors are trying to do, but it is worth thinking about and maybe even mentioning in the discussion briefly.”

Response

We sincerely appreciate the reviewer's insightful comment. We would like to clarify the possibility of selective bias arising from our genetic tagging method using the triple transgenic mouse line (pNestin-CreERT2/cfos-tTA/TRE-LSL-GCaMP6s mice) to examine adult-born neuron (ABN) activity.

Rationale for the triple-transgenic approach: The primary goal of our study was to track ABN activity in response to contextual stimuli and subsequent memory consolidation processes over several days. Achieving this required a genetically encoded Ca^{2+} indicator capable of stable and sensitive long-term imaging. We previously used GCaMP3 (Kumar et al., *Neuron*, 2020), but its dynamic range was insufficient for the prolonged observation periods required by the experimental paradigm in this study. Direct expression of newer GCaMP variants (e.g., GCaMP6s)

specifically in ABNs is also technically challenging. Although previous studies (Danielson et al., *Neuron*, 2016; Sparks et al., *Nat Comm*, 2020) achieved this through two-photon microscopy, our experimental setup required miniscope-based single-photon imaging during natural REM sleep in freely moving mice. Furthermore, expression of newer GCaMP variants in adult neural progenitor cells using pNestin-CreERT2 mice results in impaired GCaMP expression upon their maturation (Kumar et al., *Neuron*, 2020; Carrier-Ruiz et al., *STAR Protocols*, 2021). Therefore, we developed a triple transgenic system (pNestin-CreERT2/cfos-tTA/TRE-LSL-GCaMP6s) leveraging the cfos promoter, which activates specifically in mature, functional ABNs (Kee et al., *Nat Neurosci* 2007; Stone et al., *Hippocampus*, 2011).

Clarification on ABN maturation and labeling period: In our approach, recombination of the TRE-GCaMP6s gene is induced in undifferentiated nestin-positive progenitor cells by administering tamoxifen (TMX) to mice at 6 weeks of age, after which the progenitor cells are allowed 4 weeks to mature. During this period, GCaMP6s expression is initiated upon the first activation of the cfos promoter, which preferentially occurs at later stages of maturation in ABNs (Kee et al., *Nat Neurosci*. 2007; Stone et al., *Hippocampus*, 2011).

The cfos promoter is well established as a method for labeling granule cell (GN) ensembles that form contextual fear memory engrams in the dentate gyrus (DG), where ABNs are located (e.g., Liu et al., *Nature*, 2012). Moreover, a previous study demonstrated highly overlapping expression of cfos and another immediate early gene, Arc, within the DG (Stone et al., *Hippocampus*, 2011), which is also known to label contextual fear memory engrams (Denny et al., *Neuron*, 2014).

We understand the reviewer's concern that the cfos promoter in ABNs may have insufficient sensitivity to respond consistently to each contextual exposure, potentially resulting in GCaMP6s expression biased toward only a subset of ABNs. However, the experimental results from repeated contextual exposures (i.e., six sessions across 3 days) indicate that these potential issues do not substantially impact the interpretation of our findings. Specifically, if these issues had occurred, we would expect the population vector (PV) correlations among ABN ensembles to show a greater decline over time compared with those of GN ensembles. Contrary to this expectation, our data reveal consistently high stability compared with GN populations (Fig. 1i-k), which strongly suggests that our experimental approach reliably and consistently captures ABNs that are specifically responsive to the contextual stimuli used in our experiments.

It is also important to emphasize that the observed stability in PV correlations does not imply a lack of context-specificity of ABN responses. Indeed, previous studies (Kumar et al., *Neuron*, 2020; Vergara et al., *IJMS*, 2021) and additional replication experiments conducted in the current study (Extended Data Fig. 2f) clearly demonstrate that ABNs activated during contextual fear conditioning are not necessarily reactivated during memory retrieval (i.e., exposed in the same context). Furthermore, in the current study, we show that silencing ABNs responding to an

irrelevant context during REM sleep do not impact memory consolidation (Fig. 3c-d). Therefore, we consider it highly unlikely that a substantial population of ABNs functionally relevant to our memory paradigm would completely lack cfos expression. We revised the manuscript accordingly to reflect these clarifications.

Lines 215-223 (Discussion): “Regarding our method for labeling ABNs using the triple transgenic mouse line, one might raise concerns about potential bias in labeling specific subsets of neurons due to a reliance on cfos promoter activation. However, activation of the cfos promoter in DG GNs is well-established as a method for labeling cell ensembles that form contextual fear memory engrams^{17,37}. Additionally, in our repeated contextual exposure experiments, PV correlations of ABN ensembles remained stable compared with those of GN ensembles (Fig. 1i-k). These findings suggest that our approach reliably and consistently labels functionally relevant ABNs, indicating that any cell-type selection bias associated with cfos promoter activation does not substantially affect the interpretation of our results.

”

Comment 2

“For the experiments in Fig 2, when was TMX injected relative to Dox-On in the home cage? It looks like it was injected at day -14 but then the tagged ABNs would only be ~2 weeks old at the time of conditioning, and 3w old at the time of retrieval testing. This is too young to have formed synapses and make any behavioral contribution. In Fig 3 I can see that TMX was given at -4w or -10w. Can that be made more clear in Fig 2? If TMX was given at -4w (for example) in Fig 2, was Dox on or off?”

Response

In response to the reviewer’s concern, we clarified the experimental timeline in Fig. 2. Specifically, TMX was administered 4 weeks prior to the context-tagging procedure to ensure that the labeled ABNs were approximately 4 weeks old at the time of conditioning. At this age, ABNs have established synaptic connections and can meaningfully contribute to the observed behavioral outcomes.

Additionally, we revised Fig. 2 and its legend to explicitly indicate when TMX and Dox treatments occurred. We aligned these details with Fig. 3 for consistency. In particular, we now clearly mark the periods when Dox was on or off, ensuring that readers can readily understand the timing and conditions under which ABNs were labeled and tested. This improved presentation makes it evident that our experimental design ensures the ABNs’ maturity and relevance to the observed behavioral effects.

Fig. 2. “(i and k) Protocol for silencing tagged ABNs during memory retrieval. (j and l) Freezing during memory retrieval test.”

Comment 3

“I don't fully appreciate the significance of the phase-specific silencing effect. Can anything more specific be stated or speculated as to why silencing at the peak has strongest effects, or is more relevant or important?”

Response

We appreciate the opportunity to discuss this important point. Our experimental results demonstrate that suppression during the ascending phase—rather than at the peak—of theta oscillations has the most pronounced effects. This suggests that ABN ensembles are activated during this ascending phase and play a coordinated functional role with other neural circuits involved in memory consolidation during REM sleep. We explicitly address this point in the revised manuscript.

Lines 234-260 (Discussion): “In this study, we demonstrated that the activity of young ABNs at a specific phase of DG theta oscillations during REM sleep, particularly the ascending phase, is essential for the consolidation of fear memory. Periodic theta activity in the DG is generated by rhythmic inputs from the entorhinal cortex³⁷. During wakefulness, ABNs exhibit higher synchronization to theta oscillations compared with mature GNs⁹. This enhanced theta synchronization suggests that ABNs play a critical role in efficiently inducing spike timing-dependent plasticity within theta cycles. Indeed, it is well documented that the efficiency of synaptic plasticity induction in the DG depends on theta phase³⁸. Young ABNs are susceptible to synaptic plasticity on the input side, owing to their higher membrane resistance²⁸ and weaker inhibitory inputs^{39,40}. Indeed, ABN activity during REM sleep is essential for structural plasticity of their dendritic spines associated with memory consolidation¹¹.

On the output side, young ABNs exhibit distinct short-term synaptic dynamics compared with mature GNs⁴¹, suggesting their ability to provide effective input to CA3 at specific points within the theta cycle. Based on our LFP recordings near the stratum lacunosum-moleculare, we propose that granule cell layer activity is maximized around the peak following the ascending theta phase. Thus, early firing of ABNs during the ascending phase might enhance DG output at this peak timing. Specifically, high-frequency firing of ABNs could effectively depolarize CA3 neurons⁴², thereby preparing CA3 circuits to respond more robustly to subsequent mossy fiber inputs arriving. Furthermore, ABNs may play an important role in finely tuning the balance of excitation and inhibition within theta cycles³⁷ by potentially adjusting excitatory outputs through interactions with inhibitory networks, thereby facilitating efficient information processing^{9,43}

Taken together, these mechanisms suggest that REM sleep-specific synchronization of ABN activity to the ascending phase of theta oscillations effectively promotes synaptic plasticity at both input and output synapses, facilitating neural circuit reorganization critical for memory consolidation.”

Comment 4

“How confident are the authors that the neurons expressing Jaws are really adult born? The expression in the supplementary data shows cells in the middle or superficial granule cell layer, but this is not where ABNs reside. The neurons in Fig 2 are somewhat more in the expected location. Do these cells have a high input resistance as expected? How consistently did the authors find these cells in the inner granule cell layer, or near the sgz?”

Response

The only Jaws-GFP+ cells shown in the original Extended Data Figures represent developmentally born GNs labeled by AAV (originally Extended Data Fig. 3g, now 5g), not specifically targeted ABNs. By contrast, the Jaws+ neurons shown in Fig. 2 specifically illustrate ABNs labeled in the triple transgenic mice. To clarify this distinction, we explicitly note this in the figure legend.

Extended Data Fig. 5. “(f) Schematic representation of Jaws expression in DG granule neurons and virus injection. (g) Representative image of tagged DG granule neurons.”

To ensure the specificity of Jaws expression in ABNs, we employed the Nestin-CreERT2 transgenic mouse line. This line was originally developed by Dr. Amelia Eisch’s group (Lagace et al., *J Neurosci*, 2007) and was later validated by an independent research group as the most specific among three different Nestin-CreERT2 lines (Sun et al., *J Comp Neurol*, 2014). Indeed, this line is extensively used by the other prominent research laboratories (Huckleberry et al., *Neuropsychopharmacol*, 2018; Laham et al., *eLife*, 2014) and has also been rigorously validated in our previous study (Kumar et al., *Neuron*, 2020). Furthermore, the present study utilized the cfos-tTA transgenic line alongside Nestin-CreERT2 to ensure Jaws expression predominantly in mature (i.e., 4-week-old) ABNs. Previous studies report that ABNs gradually migrate toward the middle layer of the granule cell layer (GCL) upon maturation (Altman, *J Comp Neurol*, 1990; Kuhn et al., *J Neurosci*, 1996). To confirm that cfos-tTA does not affect ABN migration, we reanalyzed the location of labeled ABNs in mice not treated with doxycycline (Fig. 2c-d, no Dox group). We found that $75.4\% \pm 15.9$ (mean \pm SD) of Jaws+ ABNs

localized within the subgranular zone (SGZ) and inner GCL. This finding aligns closely with previous reports indicating that 60-80% of 4-week-old ABNs are situated in the SGZ and inner GCL (Kempermann et al., *Development*, 2003; Esposito et al., *J Neurosci*, 2005; Duan et al., *Cell*, 2007; Ibrahim et al., *Sci Rep*, 2016). These new results are shown and discussed in the revised manuscript.

Extended Data Fig. 3. “(d) ER+/Jaws+/c-fos+ mouse. Arrows, JawsGFP+ cells. (e) Distribution of c-fos-tagged ABNs in the DG (n = 207 cells from 8 mice; error bars, SEM).”

Lines 109-110 (Results): “The distribution of tagged neurons in the GL is consistent with that in previous studies^{27–30} (Extended Data Fig. 3d-e)”

In response to the reviewer’s concern, we also performed an additional experiment to measure the input resistance of Jaws+ ABNs (n = 7 cells) in acute hippocampal slices. The measured input resistance values are consistent with previous observations in 4-week-old ABNs (200-1500 MΩ; e.g., Kennedy et al., *J Neurosci*, 2024; Vyleta et al., *PLoS ONE*, 2021; Heigele et al., *Nat Neurosci*, 2016). These new data are shown in the revised manuscript.

Extended Data Fig. 3. “(g) Input resistance of Jaws+ cells (n = 7 cells from 4 mice)”

These new data strongly support the validity and specificity of our genetic labeling method for ABNs, ensuring consistency with the existing literature.

Minor comment 1

“Minor: last sentence of abstract: “Overall, this study provides mechanistic insights into how new neurons integrate into functional circuitry in the adult brain.” Does it really? It seems to focus more on the functional role of new neurons during sleep. Perhaps the authors can adjust or reword. Same for the last sentence of the discussion. It is very dramatic and seems off topic. Maybe the authors can be a little more specific with respect to how ABNs can be “therapeutically applied in cases of neuronal loss”.”

Response

We revised the manuscript as follows:

Lines 29-32 (Summary): " Collectively, our findings demonstrate that associative memory consolidation during REM sleep critically depends on both the reactivation of minimal neuronal populations and their precise coordination within specific temporal windows defined by theta oscillations."

Lines 260-261 (Discussion): "Overall, this study contributes to elucidating the mechanisms by which memory traces mature during sleep leading to the consolidation of fear memories."

Minor comment 2

“Lines 92-93: can you really say that jaws was expressed in ABNs “that were active during context exposures”? I think all you can claim with certainty is that jaws is expressed in ABNs (unless Ca²⁺ activity was visualized in these neurons, for example)”

Response

We agree with the reviewer. We revised the manuscript to more accurately reflect our experimental observations.

Lines 111-112 (Results): "As expected, Jaws-GFP expression was observed in the ABNs in nestin/cfos/jaws (cfos+) mice (Fig. 2b) but not in nestin/jaws (cfos-) control mice (Extended Data Fig. 3a-c)."

Reviewer 2

General concern about the degree of advance:

“This article is a follow up on findings presented in a *Neuron* paper published in 2020. That paper demonstrated that a small subset of adult born neurons in DG are active during post-fear conditioning REM, and that that population's activity during post-conditioning REM benefits memory consolidation. This article makes an advance upon that finding, by focusing only on the subpopulation of neurons that are active during learning. The result is as would be expected, in the affirmative - disrupting activity in this subpopulation similarly disrupts consolidation. While the techniques employed are interesting, and the original finding reported in 2020 is also interesting, the conclusions are generally the same - i.e. this doesn't really change the interpretation of the data from the 2020 paper. Aside from this general concern about the degree of advance, there are a few additional concerns, which are listed below:”

Response

We thank the reviewer for their comments. We fully recognize the importance of clearly differentiating our current findings from those previously reported by Kumar et al., *Neuron*, 2020. While our current study indeed shares strong relevance with our previous work, it offers three distinct conceptual advances, detailed explicitly below.

1. Transient contribution of ABNs to fear memory engrams. Our previous observational study (Vergara et al., *IJMS*, 2021) following Kumar et al., *Neuron*, 2020 suggests that ABNs do not form a stable memory trace cell ensemble (i.e., an engram as defined by Josselyn et al., *Science*, 2020) but instead exhibit dynamic changes (i.e., remapping) in neuronal activity during memory consolidation. These results support the notion that ABNs do not behave as stable, classical memory engram cells, suggesting instead that they play an auxiliary role in memory trace formation, addressing a critical ongoing question in the field (e.g., Anacker et al., *Neurogenesis*, 2017).

However, our previous observational study did not conclusively clarify the functional role of these neuronal populations with dynamically changing activity during memory consolidation. Thus, it still remains plausible that reduced reactivation of ABNs during memory consolidation may reflect a selective refinement process, with each remaining neuron acquiring greater functional importance.

Furthermore, Kumar et al., *Neuron*, 2020 did not definitively demonstrate that the ABNs promoting memory consolidation were identical to those activated during learning. Thus, to clarify this issue, our current study used targeted interventions that selectively manipulate neuronal ensembles activated during learning to determine their functional roles during memory consolidation and recall. This interventional approach provides clear experimental evidence for the key findings:

1. Similar ABN ensembles are consistently reactivated upon repeated exposure to the same context.
2. Reactivation of the specific ABN ensemble during REM sleep that was active during learning is necessary for memory consolidation.
3. The same ABN ensemble exhibits minimal functional importance during memory recall.

These results indicate that ABNs recruited into engrams during learning transiently form a critical part of the fear memory trace during REM sleep, without playing a significant role during subsequent recall. This aligns well with the theoretical framework proposing dynamic reorganization of memory engrams over time (Josselyn & Tonegawa, *Science*, 2020) and experimentally demonstrates the transient yet functionally crucial contribution of ABNs to memory consolidation—a clear conceptual advance that has not previously been demonstrated. To reflect this clearly, we revised our manuscript title.

Title: Transient reactivation of small ensembles of adult-born neurons during REM sleep supports memory consolidation in mice

2. Identification of the minimal ABN ensemble required for fear memory consolidation. Our study clearly demonstrates, for the first time, that the number of ABNs functionally contributing to memory consolidation is extremely small (i.e., ~3 neurons). This addresses skepticism regarding the biological significance of ABNs given their numerical scarcity (e.g., Mendez-David et al., *Hippocampus*, 2023). Our findings significantly extend our understanding of adult neurogenesis by demonstrating that even a small number of neurons can have a meaningful functional impact on behavior.

Furthermore, crucial control experiments (Fig. 3c-d, context C labeling) show no impairment of memory consolidation when silencing ABNs that were not active during learning, reinforcing the specificity of the minimal ABN ensemble in memory consolidation. Thus, our results represent an explicit and significant conceptual advance over our earlier Kumar et al., *Neuron*, 2020 findings.

3. Necessary role of ABN synchronization with theta oscillations during REM sleep for fear memory consolidation. Theta-band synchronization is particularly prominent in the DG during REM sleep and certain awake states (Montgomery et al., *J Neurosci*, 2008). Previous research established that theta oscillations during REM sleep are necessary for fear memory consolidation by optogenetically manipulating the circuits responsible for generating these oscillations (Boyce et al., *Science*, 2016). However, these prior studies did not identify the specific hippocampal neuronal populations whose theta synchronization is functionally required during REM sleep. Another study showed enhanced memory recall by artificially synchronizing DG neurons active during learning with theta oscillations during wakefulness (Rahsepar et al., *eLife*, 2023), but such artificial synchronization does not clarify the physiological necessity of theta synchronization during sleep-dependent consolidation. Additionally, McHugh et al. *Nat Neurosci*, 2022 showed

that ABNs synchronize with theta rhythms during wakefulness but did not address the functional relevance of this synchronization during wakefulness or REM sleep. Kumar et al., *Neuron*, 2020 further demonstrated that ABNs do not significantly contribute to local theta rhythms during REM sleep, raising critical questions about whether ABN synchronization with theta oscillations plays a meaningful functional role.

Therefore, in our current study, we used closed-loop optogenetic inhibition specifically targeting ABNs synchronized to local theta oscillations during REM sleep. Our findings clearly demonstrate, for the first time, that theta phase-specific synchronization of ABN activity during REM sleep is necessary for fear memory consolidation. This provides the first direct experimental evidence that specific neuronal synchronization to theta oscillations during REM sleep is functionally critical, representing another clear conceptual advance of our current work.

Major comment 1

“1) The learning paradigm employed has not been shown to produce a form of memory that requires sleep for consolidation - much less REM sleep. The importance of the findings would be more clear if it was clear that REM sleep was vital to consolidating this memory in the first place.”

Response

We deeply appreciate the reviewers' insightful comments and constructive feedback. We address their concern by citing existing literature demonstrating the dependency of our learning paradigm on REM sleep.

First, prior studies established that contextual fear conditioning is REM sleep-dependent. For instance, Ravassard et al., *Cerebral Cortex*, 2016 demonstrated that selective deprivation of REM sleep immediately following contextual fear conditioning severely impairs subsequent memory retrieval. However, interpretations of these sleep deprivation experiments face two significant challenges: (1) potential secondary effects (e.g., stress) induced by REM sleep deprivation itself and (2) the possibility of compensatory alterations in NREM sleep or wakefulness due to REM deprivation.

To address these issues, methods that manipulate specific neural circuits during REM sleep without altering REM sleep itself are particularly effective. For example, Boyce et al., *Science*, 2016 selectively inhibited hippocampal theta oscillations during REM sleep immediately after contextual fear conditioning, demonstrating impaired memory consolidation without a change in the overall amount of REM sleep. Similarly, Kumar et al., *Neuron*, 2020 demonstrated that inhibiting hippocampal ABNs specifically during REM sleep disrupts memory consolidation without affecting REM sleep. These findings strongly support the validity of using contextual fear conditioning to induce REM sleep-dependent memory consolidation.

Regarding trace fear conditioning, there is a study reporting that sleep is required for its consolidation (Chowdhury et al., *Neurobiol Learn Mem*, 2011), although the conditioned stimuli, unconditioned stimuli, and conditioned response in this prior study differ from those in the current study. We acknowledge that no prior studies directly demonstrate the necessity of REM sleep in this learning paradigm. Nevertheless, Maezono et al. *eNeuro*, 2020 used an Alzheimer's disease mouse model to demonstrate that performance in a trace fear conditioning task positively correlates with REM sleep duration during the consolidation period, suggesting the contribution of REM sleep to trace fear conditioning memory consolidation.

Our current study significantly extends this evidence by manipulating ABN activity precisely synchronized to specific phases of theta oscillations during REM sleep, without altering total REM sleep duration. We found that inhibiting ABN activity at specific theta phases selectively impairs memory consolidation (Fig. 4). This finding provides the first causal evidence supporting REM sleep's functional role in trace fear conditioning memory consolidation, which goes beyond previous correlational evidence and thus highlights our study's novelty and significance.

We incorporated the above discussion and references into the revised manuscript to more clearly and robustly convey our argument that the learning paradigms employed indeed induce REM sleep-dependent memory consolidation.

Lines 64-68 (Results): "To analyse the activity of ABN ensembles related to behavioural experiences, we employed a classical fear conditioning paradigm (Fig. 1e), which has been shown to require REM sleep for memory consolidation^{11,15,20}. This paradigm allowed us to separately analyse the encoding of a conditioned stimulus (context A) and its association with an unconditioned stimulus (foot shock)."

Line 174-176 (Results): "We employed a trace fear conditioning paradigm, known to be hippocampus- and REM sleep amount-dependent³⁹⁻⁴¹, in which a temporal gap (i.e., trace interval) is introduced between tone and shock stimuli."

Comment 2

"2) There is a concern about the conclusion on Page 4: "These results suggest that ABN ensemble activity represents the original memory more faithfully than GN ensemble activity during REM sleep within the memory consolidation period." Can this really be fairly concluded? The number of ABNs means that they form a tiny ensemble, while GNs are a significantly larger one (by orders of magnitude). So doesn't it stand to reason that with orders of magnitude more degrees of freedom for the GNs, there would almost by definition be more variability in their activity patterns across time? It isn't clear that this comparison is statistically appropriate."

Response

We appreciate the reviewer's concern regarding the difference in ensemble sizes between ABNs and GNs and whether this affects our conclusion. Below, we clarify the issue and steps we have taken to ensure our analysis is appropriate.

Imaging subsets of comparable, functionally relevant populations: Although the overall population of GNs in the DG outnumbers that of ABNs, our experiment specifically targeted cfos-tagged neurons that were activated during the learning session in both populations. This approach not only results in smaller and more comparable ensemble sizes (i.e., ABNs: $n = 42, 36,$ and 58 ; GNs: $n = 142, 125,$ and 96), but—crucially—it ensures that we focus on functionally homogeneous subsets of both ABNs and GNs. By selecting cells that express cfos in response to learning, we can more directly compare ABNs and GNs that are actively engaged in memory-related processes, thereby avoiding the confound of including large numbers of non-learning-related GNs.

Population dimensionality and heterogeneity: We acknowledge that, simply by virtue of their larger population size, GNs might be more heterogeneous and encode a broader range of cognitive processes. If this is the case, GNs would be expected to exhibit higher dimensionality than ABNs. To evaluate potential differences in the breadth or specificity of their activity patterns, we performed principal component analysis of PVs for each mouse, comparing the percentage of variance explained by each principal component between ABNs and GNs. We found no significant difference in these variance profiles, indicating that both populations have comparable dimensionality in their activity patterns. These findings are shown in the revised manuscript.

Extended Data Fig. 2. “(a) Data dimensionality between ABNs and GNs. [. . .] GNs, $n = 363$ neurons in 3 mice; ABNs, $n = 136$ neurons in 3 mice.”

Controlling for population size differences: In our original submission, we included neuron type (ABN vs. GN) as a covariate in the linear model so that we could focus on how these populations interacted with REM sleep rather than on raw differences in correlations (Fig. 1k). To address the reviewer's concerns more explicitly, we extended our model by adding the total number of neurons per mouse as an additional covariate, which helps control for potential biases introduced by

larger ensembles. We also conducted a downsampling analysis by randomly reducing the number of GNs to match the ABN count (repeated 1,000 times). Under all these conditions, the key interaction between time and REM sleep remained significant, indicating that our findings are not merely a consequence of differences in ensemble size. These findings are shown in the revised manuscript.

Extended Data Fig. 2. “(b, c [. . .]) Effect of each variable on population vector (PV) correlations when (b) the number of neurons was included as a covariate, (c) the number of GNs was downsampled to match that of ABNs [. . .].”

In light of the reviewer’s observations, we also revised the manuscript to emphasize that ABNs may be more selectively involved in the memory process without implying that GNs play a lesser role in encoding.

Lines 93-95 (Results): “These findings suggest that, as a whole, ABN ensembles exhibit a more consistent activity profile across learning and REM sleep, whereas GNs display greater heterogeneity in their involvement during these stages.”

Comment 3

“3) Related to the analysis in the point above, the conclusion about the effect sizes in REM, in particular, may be statistically spurious, given the smaller number of time points for comparison in REM vs. other states. To make this more problematic, REM is known to be suppressed after fear conditioning in mice, meaning that the number of comparisons is selectively reduced relative to other states.”

Response

We appreciate the reviewer raising an important concern regarding the statistical robustness of our REM sleep analysis findings. To address this concern, we first examined whether there was a significant difference in the total amount of sampled REM sleep between ABN and GN groups. As we found no such difference, this suggests that any potential statistical limitations due to fewer data points in REM sleep would similarly affect both groups. Additionally, to control for potential biases introduced by the reduced data points available for REM sleep relative to other sleep states, we randomly downsampled data points from the habituation and pre-/post-

imaging periods for each mouse to match the total REM sleep duration. This downsampling yielded results consistent with our original analyses. Thus, our conclusions appear robust despite the fewer REM sleep data points. We believe these additional analyses, which are shown in the revised manuscript, adequately address the reviewer's concerns and strengthen the statistical validity of our findings.

Extended Data Fig. 2. “(d) Amount of REM sleep in each group. (e) [Effect of each variable on population vector (PV) correlations when . . .] each session was downsampled to the duration of REM sleep. [. . .] GNs, $n = 363$ neurons in 3 mice; ABNs, $n = 136$ neurons in 3 mice.”

Comment 4

“4) From page 5: “Consistent with previous findings, only a few ABNs were tagged in *cfos*+ (Fig. 2b-d), with a mean (standard deviation) of 4.7(4.7) neurons per mouse in the light-effective area”. While it is clear that this population has to be small, almost by definition, how are readers to feel confident that this population even exists across all mice? Isn't it plausible that, given the SD, the *cfos*+ ABN population in some mice will actually be zero neurons?”

Response

As pointed out by the reviewer, it is indeed possible that, in some mice, ABNs expressing Jaws-YFP within the effective illumination area might not be detected. Indeed, in our original manuscript, we reported a mouse in which no Jaws-YFP+ cells were identified after exposure to Context C (original Extended Data Fig. 3c, revised Extended Data Fig, 5c). We consider several possible explanations for this phenomenon, including individual differences among mice in responsiveness to context stimulation as well as inherent sensitivity limitations of the *cfos* promoter-driven Jaws-YFP expression system.

To directly address this concern, we conducted additional immunohistochemical experiments using the same triple-transgenic mouse line (*Nestin/cfos/Jaws*). Our new analyses confirm that the average number of *cfos*+ ABNs within the effective illumination range (i.e., upper layer of the dorsal hippocampal DG) is 2.4 ± 3.2

neurons per mouse (range: 0-10 neurons, n = 8 mice; new data included in Fig. 2d). This result aligns closely with the data reported in our original manuscript, reaffirming that variability in Jaws-YFP expression likely arises due to technical limitations of the labeling approach. Importantly, the number of Jaws-YFP+ neurons identified in response to the learning stimuli in our study falls within a similar range as previous reports of *cfos*+ neuron counts (e.g., Stone et al., *Hippocampus*, 2008).

Next, we considered whether the presence of individual mice with no labeled Jaws-YFP+ cells within the illumination area would impact the overall conclusions of our study. Upon examining behavioral data from individual mice subjected to optogenetic illumination during REM sleep (Fig. 3b), we observed cases in which some mice appeared to display similar memory consolidation compared with control averages. In these cases, it is plausible that memory consolidation occurred normally due to the absence of Jaws-YFP+ neurons within the illuminated area. Ideally, correlation analyses between labeled cell counts and behavioral phenotypes would clarify this point. However, given the substantial variability in both labeling efficiency and behavioral measures (i.e., freezing), such individual-level correlation analyses are practically challenging to perform reliably.

Therefore, we adopted a rigorous alternative approach in the original manuscript: we substantially increased the number of mice per experimental group (Fig. 3; n > 12). Additionally, for the critical experiment (Fig. 3a-b), we performed an independent replication experiment with strict randomization and blinding procedures. The replication study consistently reproduced the original findings (Extended Data Fig. 5a-b).

Taken together, the original and additional analyses strongly support our primary conclusion that even a small number of ABNs play a critical role in memory consolidation, despite the possibility that some individual mice might lack detectable Jaws-YFP+ neurons within the illuminated region. We incorporated these additional data and clarifications into the main text and relevant figure (Fig. 2) to enhance the clarity and transparency of our manuscript. Control experiments, including labeling in a different context (Fig. 3c-d), further confirmed the robustness of the results.

Lines 224-233 (Results): "Some mice exhibited no detectable Jaws-GFP+ cells, likely due to individual variability in responsiveness or technical limitations of the cfos promoter-driven expression system (Fig. 2a). Similarly, within the experimental group, some mice exhibited memory consolidation comparable to control mice (Fig. 3b). Due to technical challenges preventing reliable individual-level correlation analyses, we increased the sample size (n > 12 mice per group), replicated the critical experiment with rigorous randomization and blinding (Extended Data Fig. 5a-b), and further confirmed the robustness of results through control experiments, including labeling experiments conducted in a different context (Fig. 3c-d). Overall, our analyses support the conclusion that even a very small number of ABNs can make a major contribution to memory consolidation."

Fig. 2. “(c) Dox dose- and experience-dependent analysis of Jaws-GFP expression. (d) Jaws-GFP cell density (No Dox, $n = 5$ mice; Groups 1-2, $n = 4$; Group 3, $n = 8$).”

Comment 5

“5) A very critical piece of information for interpreting all of these data is confirmation that (e.g.) Jaws-GFP expression is restricted to DG, vs present in other brain areas. Given the cropping of the images, this is an unanswered question - this it is impossible to feel confident in the behavioral results.”

Response

We thank the reviewer for their valuable comment. The pNestin-CreERT2 mouse line (Lagace et al., *J Neurosci*, 2007) used in this study is well-established for investigating hippocampal neurogenesis and has been extensively utilized in previous studies (e.g., Huckleberry, *Neuropsychopharmacol*, 2018; Sparks, *Nat Commun*, 2020), including our own work (Kumar et al., *Neuron*, 2020). In this mouse model, administration of TMX activates Cre recombinase exclusively in Nestin+ progenitor cells within the DG and the subventricular zone (SVZ), thereby inducing expression of the target gene (in this case, Jaws-GFP). By initiating gene expression in progenitor cells through TMX injection and subsequently allowing a 4-week maturation period, we can specifically study the functionality of newborn neurons at this specific age.

Previous research (Sun et al., *J Comp Neurol*, 2014) confirmed that the pNestin-CreERT2 line used in our current study achieves gene recombination specifically in Nestin+ neural progenitor cells with superior specificity over similar mouse lines. Our own prior study using this mouse model also supports this specificity (Kumar et al., *Neuron*, 2020; Supplementary Fig. S6, relevant portion shown below for reference). Although recombined cells in the SVZ migrate via the rostral migratory stream into the olfactory bulb upon maturation, we provide new data demonstrating that expression in the olfactory bulb—where our optical stimulation does not reach—is extremely limited when using our activity-dependent Jaws-GFP expression system (cfos-tTA/TRE-LSL-Jaws-GFP, Extended Data Fig. 3f).

To achieve specificity for hippocampal ABNs, we installed an optical fiber directly above the stratum lacunosum-moleculare of the DG, ensuring that optogenetic

manipulation occurs exclusively within the region illuminated by the fiber tip. This design significantly minimizes non-specific effects on other brain regions.

Kumar et al., *Neuron*, 2020, Supplemental Data 1. “Tamoxifen injection in Nestin mice induces gene expression in dentate gyrus ABNs. (A-C) Images of *tdTomato*+ and *NeuN*+ neurons at 2 weeks (A–A’), 4 weeks (B–B’), and 10 weeks (C–C’) after tamoxifen (TAM) injection in *tdTomato*^{nestin} mice (scale bar, 30 μm). (D) Density. (E) Percentages of *NeuN*+ cells within *tdTomato*+ cells. (G) Percentages of *DCX*+ cells within *tdTomato*+ cells. (F–H) example image of *tdTomato*+/*Dcx* (F), *GCaMP3*/*Dcx*, and *HaloYFP*/*Dcx* cells (I).”

Moreover, our study compares behavioral outcomes based on the maturity of hippocampal ABNs (4-week-old vs. 10-week-old), revealing behavioral phenotypes exclusively when manipulating 4-week-old neurons (Fig. 3b). If non-specific cells (i.e., not newborn neurons) contribute significantly to the observed behavioral phenotype, explaining this maturity-dependent result would be challenging. Our findings align with previous studies demonstrating that ABNs specifically contribute to memory consolidation at around 4 weeks of age (Kumar et al., *Neuron*, 2020) and with other studies using this genetic mouse line that similarly pinpoint the role of 4- to 6-week-old ABNs in memory and learning (Huckleberry, *Neuropsychopharmacol*, 2018; Sparks, *Nat Commun*, 2020). This consistency underscores the extremely limited nature of any off-target effects.

Additionally, we include a low-magnification image of the DG showing broader expression of Jaws-GFP in the revised manuscript, accompanied by descriptive text.

Extended Data Fig. 3. “(d) ER+/Jaws+/c-fos+ mouse. Arrows, Jaws-GFP+ cells.

Lines 111-112 (Results): “As expected, tagged neurons were also observed in the olfactory bulb (Extended Data Fig. 3f).”

Based on these considerations, we believe the specificity of hippocampal newborn neuron manipulation in our study is adequately validated.

Reviewer 3

Comment 1

“The tour de force experiments described in the manuscript, “Adult-born neuron reactivation in REM sleep for memory consolidation” by Srinivasan et al., give invaluable insights on the role of adult neurogenesis in memory and, in so doing, show fundamental mechanisms underlying sleep-dependent consolidation. Using incredibly difficult genetic tracing and optogenetic silencing to establish the necessity of ABNs during REM sleep, the authors convincingly demonstrate the presence of ABN memory ensembles (cfos+ engrams) specific to memory consolidation and distinguishing them uniquely from unrelated ensembles (cfos- engrams) using contextual fear conditioning and trace fear conditioning. It is especially surprising that just 5 ABN may be sufficient to establish the necessary engram. My main concern is related to Fig 4 where the authors propose that these ABN memory consolidation ensembles can be further defined by being phase-locked to theta cycles during REM (see below). By addressing my concerns, I believe that this manuscript will be well-suited to Nature Communications given its significant technical and conceptual advances which will be of great interest to a wide audience, especially those interested in neurogenesis, engrams, and sleep.

Major concerns:

1) It is unclear why the authors used a version of a contextual fear conditioning task involving exposure to Context A six times over three days. Based on the references they cited, the authors are aware of the studies by Denny et al (*Hippocampus*, 2012) and especially Drew et al (*Behav Neurosci*, 2010) showing that this much pre-exposure is not necessary and may in fact complicate interpretation of results. The authors should provide adequate justification for the use of this revised contextual fear conditioning protocol.”

Response

We sincerely appreciate the reviewer’s comment. We clarify below the rationale for conducting a total of six pre-exposures to the context, especially regarding the experiments shown in Fig. 1-3.

As the reviewer accurately notes, contextual fear conditioning itself can be established without prior contextual exposure, as demonstrated by previous studies such as Denny et al., *Hippocampus*, 2012 and Drew et al., *Behav Neurosci*, 2010. However, our primary research goal was not simply to establish contextual fear memory. Rather, we aimed to thoroughly investigate how the activity patterns of ABNs evolve during repeated exposures to the same context—from initial novelty through habituation. In addition, we sought to clearly distinguish the different phases involved: (1) contextual learning (conditioned stimulus (CS)) alone, (2) subsequent formation of an association between the CS and unconditioned stimulus (i.e., foot

shock), and (3) memory consolidation during REM sleep following the associative learning event.

Indeed, prior studies by Sahay et al., *Nature*, 2011 and Danielson et al., *Neuron*, 2016 demonstrate that multiple exposures to the same context are critical for ABNs to encode contextual details and refine fear memories. Based on these findings, we reasoned that multiple contextual exposures would be essential to enable ABNs to accurately encode detailed contextual information.

Indeed, our repeated context exposure paradigm revealed a novel finding: contextual representations by ABNs were more stable over multiple exposures compared with those by developmentally born GNs. Notably, this observation extends the previous findings of Danielson et al., *Neuron*, 2016—who examined ABN dynamics during a running task—to a purely contextual, non-running condition, underscoring the conceptual advance provided by our study.

The studies highlighted by the reviewer, such as Denny et al., *Hippocampus*, 2012 and particularly Drew et al., *Behav Neurosci*, 2010, indeed suggest that prior context exposures may reduce the involvement of neurogenesis in memory encoding. However, it is important to note that these previous studies manipulated adult neurogenesis broadly, affecting the entire process of neurogenesis itself. By contrast, our study uniquely targets a specific subpopulation of 4-week-old ABNs that were activated during a defined learning experience, specifically examining their reactivation during REM sleep and their selective contribution to memory consolidation. Thus, our use of repeated contextual exposure is fully consistent with the existing literature and effectively addresses distinct questions concerning ABN function. We concisely discuss these considerations in the revised manuscript.

Lines 64-68 (Results): “To analyse the activity of ABN ensembles related to behavioural experiences, we employed a classical fear conditioning paradigm (Fig. 1e) [. . .]. This paradigm allowed us to separately analyse the encoding of a conditioned stimulus (context A) and its association with an unconditioned stimulus (foot shock).”

Comment 2

“2) Fig. 4 – the authors need to clarify why was trace fear conditioning included in their studies. It seems unnecessary given their strong use of contextual fear conditioning in previous figures. Including trace fear conditioning puts into question whether their phase-locking results are only possible with this behavior and not the contextual fear conditioning paradigm. The authors therefore will need to provide adequate justification or provide additional experiments with contextual fear conditioning only and contrast results with trace fear conditioning.”

Response

Our previous studies demonstrate that the activity of ABNs during REM sleep is essential for consolidating contextual fear memory (Kumar et al., *Neuron*, 2020), which is a hippocampus-dependent memory task (Kim & Fanselow, *Science*, 1992). However, it remained unclear whether ABN activity during REM sleep similarly contributes to the consolidation of other types of hippocampus-dependent memories. To address this question, we used trace fear conditioning, which allows us to simultaneously examine another hippocampus-dependent memory modality—trace fear memory (McEchron et al., *J Neurosci*, 1998; Kitamura et al., *Cell*, 2014)—alongside contextual fear conditioning.

We selected the trace fear conditioning paradigm for two reasons. First, previous research (Kumar et al., *Neuron*, 2020) using a hippocampus-independent delayed tone-fear conditioning paradigm demonstrates that ABN activity during REM sleep plays little role in consolidating tone fear memory that is not hippocampus-dependent (Kim & Fanselow, *Science*, 1992). By employing the trace fear conditioning paradigm, which introduces a temporal gap between tone and shock while maintaining the tone stimulus, we can more clearly evaluate the contribution of ABNs to memory consolidation and significantly minimize confounding factors in our experimental paradigm. Second, the trace fear conditioning paradigm enables the simultaneous analysis of contextual fear memory consolidation within the same paradigm.

Through this approach, we revealed that ABN activity synchronized to specific theta phase oscillations during REM sleep is critically involved in the consolidation of not only contextual fear memory but also trace fear memory. We revised the manuscript to clarify our rationale.

Lines 174-183 (Results): “We employed a trace fear conditioning paradigm, known to be hippocampus-dependent^{33,34}, in which a temporal gap (i.e., trace interval) is introduced between tone and shock stimuli. This paradigm allowed us to simultaneously assess the involvement of ABNs in both trace fear memory and contextual fear memory consolidation within a single experimental framework. Previously, we used a delayed tone fear conditioning paradigm and demonstrated that ABN activity during REM sleep plays a minimal role in the consolidation of delayed tone fear memory¹¹, which is hippocampus-independent³⁵. Given that the only essential difference between trace fear conditioning and delayed tone fear conditioning is the inclusion of a trace interval, this paradigm is particularly advantageous for evaluating hippocampus-dependent roles of ABNs.”

Comment 3

“3) Fig. 4 – the authors compared Phase 1, 2, 3, and 4 only to the yoke controls. They need to show statistical analysis between phases in order to make the argument that ABN memory consolidation ensembles are locked to a specific phase of the theta cycle. For instance, in Fig. 4e (context test), it appears that Phase 1 may be significantly different than Phase 3 and 4 but possibly not Phase 2. However, in

Fig 4e (trace test), it appears that there is no significant difference between phases. If true, then together these results may even strengthen the authors' argument. If not, then the authors will have to more accurately qualify their conclusions on phase-locking."

Response

We intentionally refrained from conducting exhaustive post-hoc comparisons between the phase groups and yoked controls in Fig. 4e for several reasons. As shown in Fig. 4d, precise stimulation at a single theta phase was not feasible, as this led to overlaps in stimulation timing across groups, making the data non-independent and introducing collinearity. In addition, performing exhaustive post-hoc tests would substantially reduce statistical power, limiting our ability to detect reliable differences.

To more robustly assess whether memory consolidation is modulated by theta phase, we used an alternative analysis presented in Extended Data Fig. 8, which models freezing behavior as a continuous function of stimulation timing across the theta cycle. Rather than relying on the intended (i.e., targeted) stimulation phase of each group, this approach used the actual light delivery patterns observed in each mouse, which included off-target stimulations, allowing us to assess behavioral effects across the full theta cycle.

In the data shown in Extended Data Fig. 8, we applied ridge regression, a form of linear regression particularly well-suited for situations where predictors—here, neighboring theta phases—are highly correlated. Ridge regression introduces a penalty term, λ , which reduces the influence of unstable or redundant predictors. This regularization improves model stability and prevents overfitting. The optimal value of λ was determined through cross-validation to ensure generalizable results.

This analysis revealed the clear phase-dependent modulation of freezing behavior, providing strong support for our conclusion that memory consolidation is influenced by theta phase. Notably, the effect was more pronounced in the context test than in the trace test, consistent with the reviewer's observation. Taken together, we believe this modeling approach is not only necessary given the nature of the data but also more appropriate and statistically robust than post-hoc group comparisons for evaluating phase-locking effects during REM sleep.

Minor comment 1

"Minor concerns:

1) Line 86 – the authors indicate that ABN ensemble activity more faithfully represents the original memory than GN. The authors need to clarify what they mean by "more faithfully". Could the more stable PV they calculated for ABN possibly be due to their higher baseline activity levels than GN, and does not necessarily mean that memories are more faithfully encoded in ABNs over GNs?"

Response

We agree that higher baseline activity levels in ABNs could potentially explain a more stable PV. However, we found no statistical difference in baseline activity levels between cfos-expressing ABNs and GNs (Fig. 1f-g). This indicates that the specific subpopulations of ABNs and GNs examined are relatively homogeneous in their overall activity levels, contrasting with what might be expected when considering the entire ABN and GN populations.

To avoid any ambiguity, we replaced the phrase ‘more faithfully’ with a more precise explanation as written below.

Lines 93-95 (Results): “These findings suggest that, as a whole, ABN ensembles exhibit a more consistent activity profile across learning and REM sleep, whereas GNs display greater heterogeneity in their involvement during these stages.”

Minor comment 2

“2) Line 119 – the authors state that there is low overlap between conditioning and retrieval ABN populations and cite their previous work. It would be helpful to readers not aware of their past studies to show actual results based of their current experiments, even if in the Supplement.”

Response

We sincerely appreciate the reviewer’s valuable comment. In response, we conducted additional analyses to clarify the overlap between ABNs activated during conditioning and those activated during retrieval. The new data confirm results similar to those reported in our previous studies using different transgenic mouse lines (Kumar et al., *Neuron*, 2020; Vergara et al., *IJMS*, 2021, demonstrating a limited overlap between ABN populations activated during conditioning and retrieval. We present these data in the revised manuscript.

Extended Data Fig. 2. “(f) Effect size for shock and retrieval. For all panels: GNs, n = 363 neurons in 3 mice; ABNs, n = 136 neurons in 3 mice.”

Lines 131-135 (Results): “Considering the low overlap between conditioning- and retrieval-activated ABN populations^{11,24} (Extended Data Fig. 2f), the reactivation of ABN ensembles representing contextual memory may not play a critical role in memory retrieval, despite the necessity of the overall activity of the ABN population^{8,27}.”

Minor comment 3

“3) Line 205-206 – the authors claim that they provide new insight into how new neurons can be therapeutically applied in cases of devastating neuronal loss. I am not so sure how their studies show this. It would be best to delete this part.”

Response

We initially included this statement because we believe that elucidating the role of ABNs in memory consolidation could potentially inform future therapeutic strategies targeting ABNs in neurodegenerative diseases or conditions involving neuronal loss. However, as the reviewer correctly notes, our study does not provide direct experimental validation of such therapeutic applications. Therefore, we removed this statement from the revised manuscript.

Minor comment 4

“4) The title of the manuscript is a little wonky and does not appear to encapsulate the exciting results. May I suggest changing it to something like: “Reactivation of adult-born neuronal ensembles is necessary for memory consolidation during REM sleep”. Of course, this is merely a suggestion and does not affect my enthusiasm for the manuscript.”

Response

We sincerely appreciate the reviewer’s valuable feedback and kind evaluation of our study. One key finding from our study is that ABNs do not constitute a permanent component of long-term memory engrams. Rather, they transiently reactivate during REM sleep, critically contributing to memory consolidation processes. We revised the title to more accurately reflect this essential finding.

Title: "Transient reactivation of small ensembles of adult-born neurons during REM sleep supports memory consolidation in mice"

Reference list:

- Altman J, Bayer SA. 1990. Migration and distribution of two populations of hippocampal granule cell precursors during the perinatal and postnatal periods. *J Comp Neurol* **301**:365–381. doi:10.1002/cne.903010304
- Anacker C, Hen R. 2017. Adult hippocampal neurogenesis and cognitive flexibility — linking memory and mood. *Nat Rev Neurosci* **18**:335–346. doi:10.1038/nrn.2017.45
- Boyce R, Glasgow SD, Williams S, Adamantidis A. 2016. Causal evidence for the role of REM sleep theta rhythm in contextual memory consolidation. *Science* **352**:812–816. doi:10.1126/science.aad5252
- Carrier-Ruiz A, Sugaya Y, Kumar D, Vergara P, Koyanagi I, Srinivasan S, Naoi T, Kano M, Sakaguchi M. 2021. Calcium imaging of adult-born neurons in freely moving mice. *STAR Protocols* **2**:100238. doi:10.1016/j.xpro.2020.100238
- Chowdhury A, Chandra R, Jha SK. 2011. Total sleep deprivation impairs the encoding of trace-conditioned memory in the rat. *Neurobiology of Learning and Memory* **95**:355–360. doi:10.1016/j.nlm.2011.01.009
- Danielson NB, Kaifosh P, Zaremba JD, Lovett-Barron M, Tsai J, Denny CA, Balough EM, Goldberg AR, Drew LJ, Hen R, Losonczy A, Kheirbek MA. 2016. Distinct Contribution of Adult-Born Hippocampal Granule Cells to Context Encoding. *Neuron* **90**:101–112. doi:10.1016/j.neuron.2016.02.019
- Denny CA, Burghardt NS, Schachter DM, Hen R, Drew MR. 2012. 4- to 6-week-old adult-born hippocampal neurons influence novelty-evoked exploration and contextual fear conditioning. *Hippocampus* **22**:1188–1201. doi:10.1002/hipo.20964
- Denny CA, Kheirbek MA, Alba EL, Tanaka KF, Brachman RA, Laughman KB, Tomm NK, Turi GF, Losonczy A, Hen R. 2014. Hippocampal Memory Traces Are Differentially Modulated by Experience, Time, and Adult Neurogenesis. *Neuron* **83**:189–201. doi:10.1016/j.neuron.2014.05.018
- Drew MR, Denny CA, Hen R. 2010. Arrest of adult hippocampal neurogenesis in mice impairs single- but not multiple-trial contextual fear conditioning. *Behavioral Neuroscience* **124**:446–454. doi:10.1037/a0020081
- Duan X, Chang JH, Ge S, Faulkner RL, Kim JY, Kitabatake Y, Liu X, Yang C-H, Jordan JD, Ma DK, Liu CY, Ganesan S, Cheng H-J, Ming G, Lu B, Song H. 2007. Disrupted-In-Schizophrenia 1 Regulates Integration of Newly Generated Neurons in the Adult Brain. *Cell* **130**:1146–1158. doi:10.1016/j.cell.2007.07.010
- Espósito MS, Piatti VC, Laplagne DA, Morgenstern NA, Ferrari CC, Pitossi FJ, Schinder AF. 2005. Neuronal Differentiation in the Adult Hippocampus Recapitulates Embryonic Development. *J Neurosci* **25**:10074–10086. doi:10.1523/JNEUROSCI.3114-05.2005
- Heigele S, Sultan S, Toni N, Bischofberger J. 2016. Bidirectional GABAergic control of action potential firing in newborn hippocampal granule cells. *Nat Neurosci* **19**:263–270. doi:10.1038/nn.4218
- Huckleberry KA, Shue F, Copeland T, Chitwood RA, Yin W, Drew MR. 2018. Dorsal and ventral hippocampal adult-born neurons contribute to context fear memory. *Neuropsychopharmacol* **43**:2487–2496. doi:10.1038/s41386-018-0109-6

- Ibrahim S, Hu W, Wang X, Gao X, He C, Chen J. 2016. Traumatic Brain Injury Causes Aberrant Migration of Adult-Born Neurons in the Hippocampus. *Sci Rep* **6**:21793. doi:10.1038/srep21793
- Josselyn SA, Tonegawa S. 2020. Memory engrams: Recalling the past and imagining the future. *Science* **367**:eaaw4325. doi:10.1126/science.aaw4325
- Kee N, Teixeira CM, Wang AH, Frankland PW. 2007. Preferential incorporation of adult-generated granule cells into spatial memory networks in the dentate gyrus. *Nat Neurosci* **10**:355–362. doi:10.1038/nn1847
- Kempermann G, Gast D, Kronenberg G, Yamaguchi M, Gage FH. 2003. Early determination and long-term persistence of adult-generated new neurons in the hippocampus of mice. *Development* **130**:391–399. doi:10.1242/dev.00203
- Kennedy WM, Gonzalez JC, Lee H, Wadiche JI, Overstreet-Wadiche L. 2024. T-Type Ca²⁺ Channels Mediate a Critical Period of Plasticity in Adult-Born Granule Cells. *J Neurosci* **44**. doi:10.1523/JNEUROSCI.1503-23.2024
- Kim JJ, Fanselow MS. 1992. Modality-Specific Retrograde Amnesia of Fear. *Science* **256**:675–677. doi:10.1126/science.1585183
- Kitamura T, Pignatelli M, Suh J, Kohara K, Yoshiki A, Abe K, Tonegawa S. 2014. Island Cells Control Temporal Association Memory. *Science* **343**:896–901. doi:10.1126/science.1244634
- Kuhn HG, Dickinson-Anson H, Gage FH. 1996. Neurogenesis in the dentate gyrus of the adult rat: age-related decrease of neuronal progenitor proliferation. *J Neurosci* **16**:2027–2033. doi:10.1523/JNEUROSCI.16-06-02027.1996
- Kumar D, Koyanagi I, Carrier-Ruiz A, Vergara P, Srinivasan S, Sugaya Y, Kasuya M, Yu T-S, Vogt KE, Muratani M, Ohnishi T, Singh S, Teixeira CM, Chérasse Y, Naoi T, Wang S-H, Nondhalee P, Osman BAH, Kaneko N, Sawamoto K, Kernie SG, Sakurai T, McHugh TJ, Kano M, Yanagisawa M, Sakaguchi M. 2020. Sparse Activity of Hippocampal Adult-Born Neurons during REM Sleep Is Necessary for Memory Consolidation. *Neuron* **107**:552-565.e10. doi:10.1016/j.neuron.2020.05.008
- Lagace DC, Whitman MC, Noonan MA, Ables JL, DeCarolis NA, Arguello AA, Donovan MH, Fischer SJ, Farnbauch LA, Beech RD, DiLeone RJ, Greer CA, Mandyam CD, Eisch AJ. 2007. Dynamic contribution of nestin-expressing stem cells to adult neurogenesis. *J Neurosci* **27**:12623–12629. doi:10.1523/JNEUROSCI.3812-07.2007
- Laham BJ, Gore IR, Brown CJ, Gould E. 2024. Adult-born granule cells modulate CA2 network activity during retrieval of developmental memories of the mother. *eLife* **12**:RP90600. doi:10.7554/eLife.90600
- Liu X, Ramirez S, Pang PT, Puryear CB, Govindarajan A, Deisseroth K, Tonegawa S. 2012. Optogenetic stimulation of a hippocampal engram activates fear memory recall. *Nature* **484**:381–385. doi:10.1038/nature11028
- Maezono SEB, Kanuka M, Tatsuzawa C, Morita M, Kawano T, Kashiwagi M, Nondhalee P, Sakaguchi M, Saito T, Saido TC, Hayashi Y. 2020. Progressive Changes in Sleep and Its Relations to Amyloid- β Distribution and Learning in Single App Knock-In Mice. *eNeuro* **7**. doi:10.1523/ENEURO.0093-20.2020
- McEchron MD, Bouwmeester H, Tseng W, Weiss C, Disterhoft JF. 1998. Hippocampectomy disrupts auditory trace fear conditioning and contextual fear conditioning in the rat. *Hippocampus* **8**:638–646. doi:10.1002/(SICI)1098-1063(1998)8:6<638::AID-HIPO6>3.0.CO;2-Q
- McHugh SB, Lopes-dos-Santos V, Gava GP, Hartwich K, Tam SKE, Bannerman DM, Dupret D. 2022. Adult-born dentate granule cells promote hippocampal

- population sparsity. *Nat Neurosci* **25**:1481–1491. doi:10.1038/s41593-022-01176-5
- Mendez-David I, David DJ, Deloménie C, Tritschler L, Beaulieu J-M, Colle R, Corruble E, Gardier AM, Hen R. 2023. A complex relation between levels of adult hippocampal neurogenesis and expression of the immature neuron marker doublecortin. *Hippocampus* **33**:1075–1093. doi:10.1002/hipo.23568
- Montgomery SM, Sirota A, Buzsáki G. 2008. Theta and Gamma Coordination of Hippocampal Networks during Waking and Rapid Eye Movement Sleep. *J Neurosci* **28**:6731–6741. doi:10.1523/JNEUROSCI.1227-08.2008
- Rahsepar B, Norman JF, Noueihed J, Lahner B, Quick MH, Ghaemi K, Pandya A, Fernandez FR, Ramirez S, White JA. 2023. Theta-phase-specific modulation of dentate gyrus memory neurons. *eLife* **12**:e82697. doi:10.7554/eLife.82697
- Ravassard P, Hamieh AM, Joseph MA, Fraize N, Libourel P-A, Lebarillier L, Arthaud S, Meissirel C, Touret M, Malleret G, Salin P-A. 2016. REM Sleep-Dependent Bidirectional Regulation of Hippocampal-Based Emotional Memory and LTP. *Cerebral Cortex* **26**:1488–1500. doi:10.1093/cercor/bhu310
- Sparks FT, Liao Z, Li W, Grosmark A, Soltesz I, Losonczy A. 2020. Hippocampal adult-born granule cells drive network activity in a mouse model of chronic temporal lobe epilepsy. *Nat Commun* **11**:6138. doi:10.1038/s41467-020-19969-2
- Stone SSD, Teixeira CM, Zaslavsky K, Wheeler AL, Martinez-Canabal A, Wang AH, Sakaguchi M, Lozano AM, Frankland PW. 2011. Functional convergence of developmentally and adult-generated granule cells in dentate gyrus circuits supporting hippocampus-dependent memory. *Hippocampus* **21**:1348–1362. doi:10.1002/hipo.20845
- Sun M-Y, Yetman MJ, Lee T-C, Chen Y, Jankowsky JL. 2014. Specificity and efficiency of reporter expression in adult neural progenitors vary substantially among nestin-CreERT2 lines. *Journal of Comparative Neurology* **522**:1191–1208. doi:10.1002/cne.23497
- Vergara P, Kumar D, Srinivasan S, Koyanagi I, Naoi T, Singh S, Sakaguchi M. 2021. Remapping of Adult-Born Neuron Activity during Fear Memory Consolidation in Mice. *Int J Mol Sci* **22**:2874. doi:10.3390/ijms22062874
- Vyleta NP, Snyder JS. 2021. Prolonged development of long-term potentiation at lateral entorhinal cortex synapses onto adult-born neurons. *PLOS ONE* **16**:e0253642. doi:10.1371/journal.pone.0253642

Reviewer 1

The authors have worked hard to sufficiently address all of my concerns. Accept.

Response

We would like to express our sincere gratitude for dedicating the reviewer's valuable time and expertise to reviewing our manuscript. The insightful comments and constructive suggestions were invaluable, and they have significantly enhanced the quality and clarity of our paper. We are deeply grateful for the reviewer's feedback and for the final decision to accept our work.

Reviewer 2

The authors have done a stellar job of working to respond to prior concerns brought up during review. With one exception, these concerns have been addressed. One final issue should be resolved prior to acceptance of the manuscript, in order to provide transparency to readers.

This outstanding issue, raised in the prior review, is referred to in the authors' response letter:

Comment 5

“5) A very critical piece of information for interpreting all of these data is confirmation that (e.g.) Jaws-GFP expression is restricted to DG, vs present in other brain areas. Given the cropping of the images, this is an unanswered question - this it is impossible to feel confident in the behavioral results.”

The authors claim to have included a zoomed-out view, but this is not evident in the revised files. Perhaps the original comment was unclear, but what should be presented is a VERY zoomed out view, providing imaging of not only DG, but areas outside DG - i.e., full hippocampal sections, or better yet, full coronal brain sections.

Response

We thank the reviewer for their careful re-evaluation and this crucial suggestion to enhance the transparency of our data. To address the reviewer's request, we have added a representative image covering **an entire brain hemisphere** on the coronal plane to **Extended Data Fig. 3d**. This provides the wide-field, "a VERY zoomed out view" that the reviewer suggested, showing not only the DG but also all surrounding brain areas.

We would like to note that this new wide-field image of the hemisphere was taken from a different coronal section than the one used for the adjacent high-magnification image. This is because GFP signals are susceptible to photobleaching, and the section originally used for the high-magnification view had faded after extensive imaging, making it unsuitable for this purpose. Therefore, a new, representative section was prepared and imaged.

It is important to clarify, however, that identifying GFP-positive cells in this entire hemisphere image is extremely challenging. This difficulty arises from two main factors: the inherently sparse population of neurons tagged by our method and the relatively weak native GFP signal. We did attempt to overcome the weak signal issue by amplifying it with immunohistochemistry. However, this approach proved unsuitable for our purpose. Not only did the indirect labeling method introduce some non-specific background signals, but it also tended to disproportionately amplify the signal in the cell soma over the neurites. This skewed representation would not accurately reflect the native distribution of Jaws-GFP throughout the entire neuron.

Given that our goal was to show the true cellular expression, this was not an acceptable outcome. Therefore, we concluded that direct imaging of the native fluorescence, despite its faintness, was the most faithful approach. Consequently, visualizing these few, sparsely distributed neurons in such a highly zoomed-out view remains technically difficult.

Despite this technical limitation, we remain confident that the observed effects on memory consolidation are not attributable to the manipulation of non-specific neurons outside the dentate gyrus. As we detailed in our previous response, the possibility of off-target effects is extremely low. The key points supporting the specificity of our approach are summarized below:

Genetic and Spatiotemporal Specificity: We utilized the well-established pNestin-CreERT2 mouse line, which restricts Cre recombination to Nestin-positive progenitor cells in the DG and SVZ. By inducing recombination and allowing a 4-week maturation period, our manipulation specifically targets newborn neurons of a distinct age.

Localized Optogenetic Manipulation: The optical fiber was implanted directly above the DG, ensuring that light for optogenetic manipulation was delivered locally. This approach minimizes the risk of affecting distant brain regions, such as the olfactory bulb, where our data shows that Jaws-GFP expression is extremely limited with our activity-dependent system (Extended Data Fig. 3f).

Age-Dependent Behavioral Phenotype: The memory impairment was observed exclusively when manipulating 4-week-old neurons, but not 10-week-old neurons (Fig. 3e-f). This striking age-dependent effect would be difficult to explain if the phenotype were caused by the manipulation of non-specific, mature neurons.

In summary, we believe that the addition of the entire hemisphere image, in conjunction with the multiple lines of evidence outlined above, fully addresses the reviewer's final concern and substantiates the specificity of our findings.

Reviewer 3

The authors have thoroughly addressed all my concerns. I strongly endorse the acceptance of this manuscript.

Response

We are immensely grateful for the considerable time and effort the reviewer devoted to reviewing our manuscript. The reviewer's perceptive feedback and valuable recommendations were instrumental in improving the paper's quality and clarity. We sincerely appreciate the meticulous review and the decision to accept our work for publication.